# INTERVENING TO LEARN AND COMPOSE CAUSALLY DISENTANGLED REPRESENTATIONS

## ABSTRACT

In designing generative models, it is commonly believed that in order to learn useful latent structure, we face a fundamental tension between expressivity and structure. In this paper we challenge this view by proposing a new approach to training arbitrarily expressive generative models that simultaneously learn causally disentangled concepts. This is accomplished by adding a simple decoder-only module to an existing decoder that can be arbitrarily complex. The module learns to process concept information by implicitly inverting linear representations from an encoder. Inspired by the notion of intervention in a causal model, our module selectively modifies its architecture during training, allowing it to learn a compact joint model over different contexts. We show how adding this module leads to causally disentangled representations that can be composed for out-of-distribution generation on both real and simulated data. To further validate our proposed approach, we prove a new identifiability result that extends existing work on identifying structured representations in nonlinear models.

## 1 INTRODUCTION

Generative models have transformed information processing and demonstrated remarkable capacities for creativity in a variety of tasks ranging from vision to language to audio. The success of these models has been largely driven by modular, differentiable architectures based on deep neural networks that learn useful representations for downstream tasks. Recent years have seen increased interest in understanding and exploring the representations produced by these models through evolving lines of work on structured representation learning, identifiability and interpretability, disentanglement, and causal generative models. This work is motivated in part by the desire to produce performant generative models that also capture meaningful, semantic latent spaces that enable out-of-distribution (OOD) generation under perturbations.

A key driver of this line of work is the tradeoff between flexibility and structure, or expressivity and interpretability: Conventional wisdom suggests that to learn structured, interpretable representations, model capacity must be constrained, sacrificing flexibility and expressivity. This intuition is supported by a growing body of work on nonlinear ICA (Hyvärinen & Pajunen, 1999), disentanglement (Bengio, 2013), and causal representation learning (Schölkopf et al., 2021). On the practical side, methods that have been developed to learn structured latent spaces tend to be bespoke to specific data types and models, and typically impose limitations on the flexibility and expressivity of the underlying models. Moreover, the most successful methods for learning structured representations typically impose fixed, known structure *a priori*, as opposed to learning this structure from data.

At the same time, there is also a growing body of work that suggests generative models already learn surprisingly structured latent spaces (e.g. Mikolov et al., 2013; Szegedy et al., 2013; Radford et al., 2015). This in turn suggests that existing models are already "close" to capturing useful structure, and perhaps only small modifications are needed. Since we already have performant models that achieve state-of-the-art results in generation and prediction, do we need to re-invent the wheel to achieve these goals? Our hypothesis is that we should be able to leverage the expressiveness of these models to build new models that learn latent structure from scratch. We emphasize that our goal is not to explain or interpolate the latent space of a pre-trained model, but rather to leverage known architectures to train a *new* model end-to-end from scratch.

Figure 1: Overview of our approach (see (4) for notation map). Given a black-box encoder-decoder architecture (blue), we propose to append a *context module* (red) to the decoder. (left) Instead of passing the output of the encoder directly to the decoder (blue box + blue arrow), the embeddings **e** are passed through the context module, consisting of three distinct layers. The output of this module is then passed into the decoder. (right) The model learns to compose different concepts OOD, in this case object and background colour, to values that do not appear together in the training data.

In this paper, we adopt this perspective: We start with a black-box architecture, upon which we make no assumptions, then augment this model to learn causally disentangled representations. The idea is that the black-box architecture is already known to perform well on downstream tasks (e.g. generation, prediction), and so is flexible enough to capture complex patterns in data. We then introduce a modular, end-to-end differentiable architecture for learning causally disentangled representations (Bengio, 2013; Schölkopf et al., 2021), by augmenting the original model with a simple decoder-only module that retains its existing capabilities while enabling concept identification and intervention as well as composing together multiple concept interventions for OOD generation (Figure 1). See Section 2 for more on causal disentanglement. The resulting latent structure is learned fully end-to-end, with no fixed structure imposed *a priori*. The motivation is to provide a framework for taking existing performant architectures and to train a new model that performs just as well, but with the added benefit of learning structured representations without imposing specific prior knowledge or structure. Since the model, unlike previous approaches, does not impose rigid latent structure, we theoretically validate the approach with a novel identifiability result for structured representations recovered by the model. For a detailed discussion of related work, see Appendix A.

**Contributions**   The main contribution of this work is to propose a differentiable module for causal disentanglement that can be inserted into any black-box decoder and trained through standard gradient-based methods. More specifically, we make the following additional contributions:

1. We introduce a simple *context module* that can be attached to an existing decoder to learn concept representations by splitting each concept into a tensor slice, where each slice represents an interventional context in a reduced form structural equation model (SEM). This module can be attached to any decoder and trained end-to-end. See Figure 1.

2. We evaluate the implications of this module through quantitative and qualitative experiments on OOD generation, which is distinct from and significantly more challenging than OOD reconstruction as adopted in prior work (e.g. Xu et al., 2022; Mo et al., 2024; Montero et al., 2022; Schott et al., 2022). We perform experiments on OOD generation on several benchmark datasets (3DIdent, CelebA, MNIST) as well as carefully controlled simulations.

3. To illustrate the adaptivity of our module to different architectures, we include experiments with different models (e.g. NVAE, Vahdat & Kautz, 2020) and multiple ablations that carefully control model capacity, data contexts, and hyperparameters, ensuring that any differences in performance are attributable to specific architectural modifications.

We also introduce quad, a simple simulated visual environment for evaluating causal disentanglement, compositional abilities, and OOD generation. As a matter of independent interest, we prove an identifiability result under concept interventions, and discuss how the proposed architecture can be interpreted as an approximation to this model. Due to space limitations, full technical proofs, along with detailed literature review and experimental details, can be found in the appendix.

## 2 PRELIMINARIES

Let $\mathbf{x}$ denote the observations, e.g., pixels in an image, and $\mathbf{z}$ denote hidden variables, e.g. latent variables that are to be inferred from the pixels. We are interested in training generative models, such as a variational autoencoder (VAE). A typical generative model consists of a decoder $p_\theta(\mathbf{x} \mid \mathbf{z})$ and an encoder $q_\phi(\mathbf{z} \mid \mathbf{x})$. After specifying a prior $p_\theta(\mathbf{z})$, this defines a likelihood by

$$p_\theta(\mathbf{x}) = \int p_\theta(\mathbf{x} \mid \mathbf{z}) p_\theta(\mathbf{z}) \, \mathrm{d}\mathbf{z}. \tag{1}$$

Both the decoder and encoder are specified by deep neural networks that will be trained end-to-end via standard techniques (Kingma & Welling, 2014; Rezende et al., 2014; Rezende & Mohamed, 2015). In this work, we focus on VAEs since they are often use to represent structured and semantic latent spaces, which is our focus. Unlike traditional structured VAEs, we do not impose a specific structure *a priori*.

**Causal disentanglement** Our goal is to learn distinct latent factors that can be composed together. This follows from the existing notion of causal disentanglement, which we recall informally here:

**Definition 2.1** (Bengio, 2013; Thomas et al., 2017). A *causally disentangled* latent space consists of separately controllable (intervenable) latent factors.

In particular, we want to manipulate and intervene on latent factors in a causally meaningful way; crucially this allows us to compose distinct concepts together without needing to learn the full causal graph in the latent space. This notion of disentanglement is also different from the axis-alignment notion (Kim & Mnih, 2018; Chen et al., 2018). The "causal" aspect comes from interpreting "separately controllable" to mean "able to intervene on independent causal mechanisms", as suggested by Locatello et al. (2019); see also Schölkopf et al. (2021). See Appendix B for formal details. Rather than attempt to reconstruct the full causal DAG model (as in Yang et al., 2021; Squires et al., 2023; Buchholz et al., 2023), we aim for the more practical goal of learning different interventional contexts through distributional invariances; thus our approach strikes a balance between easy-to-learn unstructured latent spaces and hard-to-learn but desirable fully structured causal DAG latent spaces.

**Measuring causal disentanglement** A standard approach to quantifying causal disentanglement is to evaluate the recovery of the latent causal graph; since our explicit goal is to avoid learning this graph, we propose a different metric that is more suitable to downstream applications. To measure causal disentanglement, we use OOD generation on novel interventions as a metric. We train models so that they never see examples of certain latent factors composed together (equivalently, multiple simultaneous interventions on distinct factors), so these examples are genuinely OOD. By withholding these examples during training and building a model that is capable of intervening on multiple factors, we can compare the quality of OOD samples generated from our model to the held-out OOD samples. In our experiments, we use the sliced Wasserstein metric (18); see Appendix D.1 for details.

**OOD reconstruction vs. OOD generation** Previous work has evaluated the OOD capabilities of generative models by using OOD reconstruction as a metric (e.g. Xu et al., 2022; Mo et al., 2024; Montero et al., 2022; Schott et al., 2022). There is a crucial difference between OOD reconstruction and OOD generation: By OOD generation, we mean the ability to conditionally generate and combine novel interventions on existing concepts. For example, suppose during training the model only sees small, red objects along with big, blue objects. At test time, we wish to generate OOD samples of big, red objects or small, blue objects that were never seen during training. The crucial difference is that while we can *always* attempt to reconstruct *any* sample (i.e. OOD or not) and evaluate its reconstruction error, it is not always possible to have the model *generate* a new OOD sample. Unless the model learns specific latent factors corresponding to concepts such as size or colour, we cannot control (i.e. intervene) the size or colour of random samples from the model.

Thus, OOD generation not only evaluates the ability of a model to compose learned concepts in new ways, but also the ability of a model to identify and capture underlying concepts of interest. Accordingly, we argue that OOD generation is more appropriate to evaluate causal disentanglement since it captures *both* the model's ability to learn concepts as distinct latent factors *as well as* compose them together in novel ways.

**Set-up and approach** Our set-up is the following: We have a black-box encoder and decoder, on which we make no assumptions other than the encoder outputs a latent code corresponding to $\mathbf{x}$. In

the notation of (1), this corresponds to $\mathbf{z}$, however, to distinguish the black-box embeddings from our model, we will denote the black-box embeddings hereafter by $\mathbf{e}$. Our plan is to work solely with the embeddings $\mathbf{e}$ and learn how to extract linear concept representations from $\mathbf{e}$ such that distinct concepts can be intervened upon and composed together to create novel, OOD samples. Crucially, we do not modify the black-box architecture in any way.

Our approach is motivated by the following incongruence: On the one hand, it is well-established that the latent spaces of generative models are typically entangled and semantically misaligned, fail to generalize OOD, and suffer from posterior collapse (which has been tied to latent variable nonidentifiability, Wang et al., 2021). On the other hand, they still learn highly structured latent spaces that can be traversed and interpolated, are "nearly" identifiable (Willetts & Paige, 2021; Reizinger et al., 2022), and represent abstract concepts linearly (Mikolov et al., 2013; Szegedy et al., 2013; Radford et al., 2015). Thus, our hypothesis is that sufficiently flexible models do capture meaningful structure internally, just not in a way that is causally interpretable or meaningful in practice. So, to learn latent structure, rather than build bespoke architectures from scratch we build off the already performant embeddings of existing architectures.

## 3 ARCHITECTURE

Our starting point is a black-box encoder-decoder architecture along with $d_c$ concepts of interest, denoted by $\mathbf{c} = (\mathbf{c}_1, \ldots, \mathbf{c}_{d_c})$. Our objectives are two-fold:

1. To learn structure between concepts from black-box embeddings $\mathbf{e}$ as linear projections $C\mathbf{e}$;

2. To compose concepts together in a single, transparent model that captures how different concepts are related.

We seek to learn these concepts in an end-to-end, differentiable manner.

A key intuition is that composition can be interpreted as a type of intervention in the latent space. This is sensible since interventions in a causal model are a type of nontrivial distribution shift. Thus, the problem takes on a causal flavour which we leverage to build our architecture. The difficulty with this from the causal modeling perspective is that encoding structural assignments and/or causal mechanisms directly into a feed-forward neural network is tricky, because edges between nodes within the same layer aren't allowed in a feed-forward network but are required for the usual DAG representation of a causal model. To bypass this, we use the *reduced form* of a causal model which has a clear representation as a bipartite directed graph, with arrows only from exogenous to endogenous variables, making it conducive to being embedded into a neural network. The tradeoff is that we do not learn a causal structure (i.e., a causal DAG), however, this enables the model to perform interventions directly in the latent space and to leverage invariances between interventional contexts.

### 3.1 OVERVIEW

Before outlining the architectural details, we provide a high-level overview of the main idea. A traditional decoder transforms the embeddings $\mathbf{e}$ into the observed variables $\mathbf{x}$, and the encoder operates in reverse by encoding $\mathbf{x}$ into $\mathbf{e}$. Thus,

$$\mathbf{x} \xrightarrow{\text{encoder}} \mathbf{e} \xrightarrow{\text{decoder}} \widehat{\mathbf{x}}.$$

Based on an extensive body of empirical work that shows concepts are linearly represented (Mikolov et al., 2013; Szegedy et al., 2013; Radford et al., 2015, see Appendix A for more discussion), we represent concepts as linear projections of the embeddings $\mathbf{e}$: Each concept $\mathbf{c}_j$ can be approximated as $\mathbf{c}_j \approx C_j\mathbf{e}$. This is modeled via a linear layer $\mathbf{c} \to \mathbf{e}$ that implicitly inverts this relationship in the decoder.

As a result, it makes sense to model the relationships between concepts with a linear SEM:

$$\mathbf{c}_j = \sum_{k=1}^{d_c} \alpha_{kj}\mathbf{c}_k + \boldsymbol{\varepsilon}_k, \quad \alpha_{kj} \in \mathbb{R}. \tag{2}$$

By reducing this SEM and solving for $\mathbf{c}$, we deduce that

$$\mathbf{c} = A_0 \varepsilon, \text{ where } \begin{cases} \mathbf{c} = (\mathbf{c}_1, \ldots, \mathbf{c}_{d_c}) \\ \varepsilon = (\varepsilon_1, \ldots, \varepsilon_{d_c}) \end{cases}. \tag{3}$$

This is known as the *reduced form* of the SEM (2). We model this with a linear layer $\varepsilon \to \mathbf{c}$, where the weights in this layer correspond to the matrix $A_0$. This layer will be used to encode the SEM between the concepts, which will be used to implement causal interventions.

*Remark* 3.1. Due to the reduced form SEM above, our approach *does not* and *cannot* model the structural causal model encoded by the $\alpha_{kj}$. What is important is that the reduced form $\mathbf{c} = A_0 \varepsilon$ still encodes causal invariances and interventions, which is enough in our setting, without directly estimating a causal graph.

In principle, since the exogenous variables $\varepsilon_j$ are independent, we could treat $\varepsilon$ as the input latent space to the generative model. Doing this, however, incurs two costs: 1) To conform to standard practice, $\varepsilon$ would have to follow an isotropic Gaussian prior, and 2) It enforces artificial constraints on the latent dimension $\dim(\varepsilon)$. For this reason, we use a second expressive layer $\mathbf{z} \to \varepsilon$ that gradually transforms $\mathbf{z} \sim \mathcal{N}(0, I)$ into $\varepsilon$. This allows for $\dim(\mathbf{z})$ to be larger and more expressive than $d_c$ (in practice, we set $\dim(\mathbf{z})$ to be a multiple of $\dim(\varepsilon)$), and for $\varepsilon$ to be potentially non-Gaussian.

The final decoder architecture can be decomposed, at a high-level, as follows (see also Figure 1):

$$\mathbf{z} \longrightarrow \varepsilon \longrightarrow \mathbf{c} \longrightarrow \mathbf{e} \longrightarrow \mathbf{x}. \tag{4}$$

Implementation details for each of these layers can be found in the next section.

## 3.2 DETAILS

We assume given a black-box encoder-decoder pair, denoted by $\mathrm{enc}(\mathbf{x})$ and $\mathrm{dec}(\mathbf{z})$, respectively. We modify the decoder by appending a *context module* to the decoder. This context module is divided into three layers: A *representation layer*, an *intervention layer*, and an *expressive layer*.

1. The first *representation layer*, as the name suggests, learns to represent concepts by implicitly inverting their linear representations $\mathbf{c}_j = C_j \mathbf{e}$ from the embeddings of the encoder. This is a linear layer between $\mathbf{c} \to \mathbf{e}$.

2. The *intervention layer* embeds these concept representations into the reduced form SEM (3). Each concept corresponds to its own context where it has been intervened upon. A key part of this layer is how it can be used to learn and enforce interventional semantics through this shared SEM. This layer corresponds to the second layer $\varepsilon \to \mathbf{c}$ in (4).

3. The *expressive layer* is used to reduce the (potentially very large) input latent dimension of independent Gaussian inputs down to a smaller space of non-Gaussian exogenous noise variables for the intervention layer, corresponding to the first layer $\mathbf{z} \to \varepsilon$ in (4). This is implemented as independent, deep MLPs that gradually reduce the dimension in each layer: If it is desirable to preserve Gaussianity for the exogenous variables, linear activations can be used in place of nonlinear activations (e.g., ReLU).

Because the intervention layer is modeled after an SEM, it is straightforward to perform latent concept interventions using the calculus of interventions in an SEM.

*Remark* 3.2. Crucially, no part of this SEM is fixed or known—everything is trained end-to-end. In particular, we do not assume a known causal DAG or even a known causal order. This stands in contrast to previous work on causal generative models (e.g. Kocaoglu et al., 2017; Yang et al., 2021).

**Concept interventions** For simplicity, assume for now that the concepts are each one-dimensional with $\dim(\mathbf{c}_j) = \dim(\varepsilon_j) = 1$; generalization to multi-dimensional concepts is straightforward and explained below. The structural coefficients $\alpha_{kj} \in \mathbb{R}$ capture direct causal effects between concepts, with $\alpha_{kj} \neq 0$ indicating the presence of an edge $\mathbf{c}_k \to \mathbf{c}_j$. An intervention on the $j$th concept requires deleting each incoming edge; i.e., setting $\alpha_{\cdot j} = 0$, updating the outgoing edges from $\varepsilon_j$, as well as replacing $\varepsilon_j$ with a new $\varepsilon'_j$, which results from updating the incoming edges to $\varepsilon_j$ in the preceding layer. Thus, during training, when we intervene on $\mathbf{c}_j$, we zero out the row $\alpha_{\cdot j}$ while replacing the column $\alpha_{\cdot j}$ with a new column $\beta_{\cdot j}$ that is trained specifically for this interventional setting. The result

is $d_c + 1$ intervention-specific layers: $A_0$ for the observational setting, and $A_1, \ldots, A_{d_c}$ for each interventional setting. This is captured as a three-way tensor where each slice of this tensor captures a different context that corresponds to intervening on different concepts. At inference time, to generate interventional samples we simply swap out the $A_0$ for $A_j$. Moreover, multi-target interventions can be sampled by zeroing out multiple rows and substituting multiple $\beta_j$.'s and $\varepsilon_j$'s.

**Multi-dimensional concepts** Everything above goes through if we allow additional flexibility with $\dim(\mathbf{c}_j) > 1$ and $\dim(\varepsilon_j) > 1$, potentially even with $\dim(\mathbf{c}_j) \neq \dim(\varepsilon_j)$. In practice, we implement this via two width parameters $w_\varepsilon = \dim(\varepsilon_j)$ and $w_c = \dim(\mathbf{c}_j)$. The design choice of enforcing uniform dimensionality is not necessary, but is made here since in our experiments there did not seem to be substantial advantages to choosing nonuniform widths. With these modifications, $\alpha_{kj}$ becomes a $w_\varepsilon \times w_c$ matrix; instead of substituting rows and columns above, we now substitute slices in the obvious way. The three-way tensor $A$ is now $(d_c + 1) \times w_\varepsilon d_c \times w_c d_c$-dimensional.

**Identifiability** An appealing aspect of this architecture is that under certain assumptions, it can be viewed as an approximation to an identifiable model over disentangled concepts. Formally, we assume given observations $\mathbf{x}$, concepts $\mathbf{c}$, and a collection of embeddings $\mathbf{e}$ such that $\mathbf{x} = f(\mathbf{e})$. Additionally assume a noisy version of the linear representation hypothesis, i.e., $\mathbf{c}_j = C_j \mathbf{e} + \varepsilon_j$ with $\varepsilon_j \sim \mathcal{N}(0, \Omega_j)$. Then we have the following identifiability result:

**Theorem 3.1** (Identifiability). *Assume that the rows of each $C_j$ come from a linearly independent set and $f$ is injective and differentiable. Then, given single-node interventions on each concept $\mathbf{c}_j$, the representations $C_j$ and latent concept distribution $p(\mathbf{c})$ are identifiable.*

For a formal statement, see Theorem B.1 in Appendix B. As opposed to solving a causal representation learning problem, the causal semantics used in this paper are simply being ported into a generative model to give solutions to the problem of learning latent *distribution* structure, as in Theorem 3.1. The technical difficulty in the proof of Theorem 3.1 involves analyzing interventions in this model; since our focus is on implementation and practical aspects, we defer further discussion to Appendix B.

### 3.3 SUMMARY OF ARCHITECTURE

The context module thus proposed offers the following appealing desiderata in practice:

1. **Black-box.** There is no coupling between the embedding dimension and the number of concepts, or the complexity of the SEM. In particular, this architecture allows for *arbitrary* black-box encoder-decoder architectures to be used to learn embeddings, from which a causal model is then trained on top of.

2. **Causality.** The intervention layer is a genuine causal model that provides rigorous causal semantics that allow sampling from arbitrary concept interventions, including interventions that have not been seen during training.

3. **Identifiability.** The architecture is based on an approximation to an identifiable concept model (Theorem 3.1), which provides formal justification for the intervention layer as well as reproducibility assurances.

4. **Flexibility.** The context module is itself arbitrarily flexible, meaning that there is no risk of information loss in representing concepts with the embeddings $\mathbf{e}$. Of course, information loss is possible if we compress this layer too much (e.g., by choosing $d_c$, $k$, $w_\varepsilon$, or $w_c$ too small), but this is a design choice and not a constraint of the architecture itself.

As a consequence, the only tradeoff between representational capacity and causal semantics is design-based: The architecture itself imposes no constraints. The causal model can be arbitrarily flexible and chosen independent of the black-box encoder-decoder pair, which is also allowed to be arbitrary.

## 4 EMPIRICAL EVALUATION

We evaluate the proposed context module on three different tasks:

1. **Concept learning** (Section 4.1): *How well does the context module learn representations that correspond to each concept?* To measure this, we generate interventional samples by intervening on a single concept, and directly compare these interventional samples to held-out samples using the SW metric (18).

2. **Composition** (Section 4.2): *How well does the context module learn to* compose *these concepts together in novel ways that have not been seen during training?* To measure this, we generate unseen samples by intervening on multiple concepts, and directly compare these interventional samples to held-out samples using the SW metric (18). By design, interventions on multiple concepts are never seen during training, so this is genuinely OOD.

3. **Reconstruction** (Section 4.3): *How well does the context module reconstruct samples from each context?* This can be evaluated using standard measures of generation quality; we use bits per dimension (bpd) for comparison with prior work. Since this is computed across all contexts, this captures the model's ability to reconstruct samples from diverse contexts.

For concept learning and composition, including OOD generation (Section 2), we report the sliced Wasserstein (SW) distance (Bonneel et al., 2015; Flamary et al., 2021) between a held-out sample from the ground-truth distribution and a sample generated (not reconstructed) by the trained model. For reconstruction, we report the standard ELBO loss (Kingma & Welling, 2014) in bits per dimension. We evaluate our approach by augmenting two base architectures—a lightweight convolutional $\beta$-VAE (Higgins et al., 2017) and a state-of-the-art deep hierarchical network, NVAE (Vahdat & Kautz, 2020)—with our context module (CM), yielding **CM+$\beta$VAE** and **CM+NVAE**. This allows us to compare to the base architectures without the context module, as well as ablations with **pooled** (pooling all contexts into a single context) and observational (**obs**, omitting interventional contexts) data. This allows us to distinguish performance differences due to data diversity and/or architectural complexity. For more details and complete results, see Appendices D-E.

We compare these models to our context module on 3DIdent (Zimmermann et al., 2021), Morpho MNIST (Castro et al., 2019), and CelebA (Liu et al., 2015). In order to carefully evaluate model performance in a controlled environment over different random seeds (and hence assess statistical variability), we also introduce quad, a novel semi-synthetic visual environment with intervenable latents (colour, shape, size, orientation) that explicitly enables sampling from ground-truth OOD composed contexts. See Appendix C for more details on this dataset.

## 4.1 CONCEPT LEARNING

Tables 1 and 2, in the **Concept learning** (top) rows, show a quantitative evaluation of concept learning on the quad and 3DIdent datasets. Both of these datasets enable sampling genuinely OOD contexts by intervening on concepts, making it easy to measure OOD generation quality in Section 4.2. We draw three main conclusions from these tables, by comparing the context module + base method to each of the other three methods (columns).

First, comparing the **CM+NVAE** column to the **pooled NVAE** column of Table 2, we see that despite the base NVAE achieving better average reconstruction performance in Section 4.3, the context module here outperforms NVAE at learning individual concepts, achieving an average Wasserstein distance of 0.018 compared to NVAE's 0.041. Examining Table 1 leads to the analogous conclusion that CM+$\beta$VAE outperforms the base $\beta$-VAE at concept learning across quad concepts—in this case the simpler model/data allows more replicates to be run, resulting in greater certainty due to a much smaller $p$-value ($< 10^{-12}$).

Second, comparing the **CM+NVAE** column to the **obs NVAE** column of Table 2, in particular looking at the **obs** row, we see that CM+NVAE is better at learning the observational distribution than an NVAE model trained only on the observational (rather than pooled) data. This indicates that the context module is able to leverage

Table 1: Results on SW metric (lower is better) on concept learning and composition performance across quad contexts. Last two rows show mean SW values and $p$-values comparing CM+$\beta$VAE to other methods.

| | | **Method** | | | |
|---|---|---|---|---|---|
| | | **CM+$\beta$VAE** | **pooled CM** | **obs $\beta$VAE** | **pooled $\beta$VAE** |
| **Concept learning** | obs | **0.032** | 0.084 | 0.043 | 0.087 |
| | orient | **0.036** | 0.082 | 0.045 | 0.086 |
| | quad1 | **0.045** | 0.257 | 0.241 | 0.209 |
| | quad2 | **0.047** | 0.240 | 0.247 | 0.223 |
| | quad3 | **0.046** | 0.243 | 0.230 | 0.220 |
| | quad4 | **0.039** | 0.238 | 0.238 | 0.210 |
| | size | **0.059** | 0.164 | 0.150 | 0.113 |
| | mean | **0.0435** | 0.1868 | 0.1703 | 0.1640 |
| | $p$-value | – | $< 10^{-13}$ | $< 10^{-9}$ | $< 10^{-12}$ |
| **Composition** | (q1, or) | **0.081** | 0.253 | 0.248 | 0.226 |
| | (q1, q2) | **0.145** | 0.363 | 0.300 | 0.284 |
| | (q1, q3) | **0.147** | 0.327 | 0.343 | 0.305 |
| | (q1, q4) | **0.097** | 0.324 | 0.337 | 0.299 |
| | (q1, sz) | **0.142** | 0.257 | 0.259 | 0.222 |
| | (q2, or) | **0.074** | 0.236 | 0.212 | 0.209 |
| | (q2, q3) | **0.162** | 0.347 | 0.361 | 0.311 |
| | (q2, q4) | **0.139** | 0.305 | 0.324 | 0.293 |
| | (q2, sz) | **0.129** | 0.274 | 0.274 | 0.205 |
| | mean | **0.1240** | 0.2985 | 0.2953 | 0.2615 |
| | $p$-value | — | $< 10^{-22}$ | $< 10^{-22}$ | $< 10^{-17}$ |

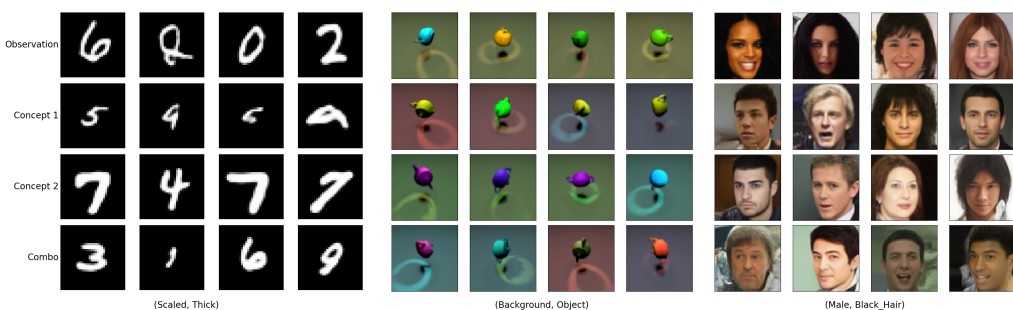

Figure 2: Examples of concept learning and composition in MNIST (left), 3DIdent (middle), and CelebA (right) using CM+NVAE, our context module augmenting the base NVAE—these images are *generated*, not reconstructed, by CM+NVAE. The first row shows generated observational samples; The second and third rows show samples of *learned concepts*, generated by intervening on individual concepts, e.g., 'Concept 1' is 'scaled' for MNIST; The final row shows OOD *composition*, generated by simultaneously intervening on pairs of learned concepts.

invariances across the different contexts, providing it with performance and computational resource advantages compared to models trained independently on the different contexts. Again, Table 1 demonstrates the same patterns, leading to the same conclusions, but with greater certainty due to the much smaller $p$-values.

Third, comparing the **CM+NVAE** column to the ablation in **pooled CM**—which uses pooled data to circumvent the context module's intervention mechanism while leaving the rest of the module architecture intact—we see that CM+NVAE consistently outperforms it. This demonstrates that the increased performance of CM+NVAE is due our module's intervention mechanism and its resulting ability to better leverage context-separated data, as opposed to spuriously benefiting from a different architecture than the baselines. Again, Table 1 demonstrates the same patterns, leading to the same conclusions, but with greater certainty due to the much smaller $p$-values.

Table 2: Results on SW metric (lower is better) on concept learning and composition performance across 3DIdent contexts. Last two rows show mean SW values and $p$-values comparing CM+NVAE to other methods.

|  |  | Method | | | |
|---|---|---|---|---|---|
|  |  | **CM+NVAE** | **pooled CM** | **obs NVAE** | **pooled NVAE** |
| **Concept learning** | **obs** | **0.019** | 0.024 | 0.053 | 0.028 |
|  | **object** | **0.017** | 0.065 | 0.089 | 0.058 |
|  | **background** | **0.018** | 0.059 | 0.066 | 0.053 |
|  | **spotlight** | **0.018** | 0.022 | 0.045 | 0.025 |
|  | **mean** | **0.018** | 0.043 | 0.063 | 0.041 |
|  | **$p$-value** | — | 0.059 | 0.009 | 0.038 |
| **Composition** | **(obj, bg)** | **0.042** | 0.075 | 0.085 | 0.068 |
|  | **(bg, sl)** | **0.046** | 0.069 | 0.095 | 0.067 |
|  | **(obj, sl)** | **0.056** | 0.073 | 0.083 | 0.063 |
|  | **mean** | **0.048** | 0.072 | 0.088 | 0.066 |
|  | **$p$-value** | — | 0.018 | 0.012 | 0.046 |

Finally, for a qualitative evaluation, we look at Figure 2, in the Concept 1 and Concept 2 rows, to see concept learning in CM+NVAE across three different datasets. For example, looking at the 3DIdent results in Figure 2 (middle), the Observation row shows the learned observational distribution of images to contain background colours and object colours both in the orange–green range (███████). The Concept 1 row, produced by intervening on the learned concept of background colour, shows that the background colour is shifted to the blue–red range while other features, like object colour, remain invariant with the observational samples. Analogously, the Concept 2 row shows a shift of object colour to the blue–red range (███████) while the other features remain invariant. These illustrate that CM+NVAE successfully learned the concepts of background colour and object colour, in line with the notion of causal disentanglement in Definition 2.1.

## 4.2 CONCEPT COMPOSITION

Tables 1 and 2, in the **Composition** (bottom) rows, provide a quantitative evaluation of concept composition on the quad and 3DIdent datasets, respectively, while the Combo row of Figure 2 provides a qualitative evaluation across the MNIST, 3DIdent, and CelebA datasets. In each evaluation, the model uses the intervention layer (Section 3.2) to intervene on pairs of learned concepts, i.e.

to compose two concepts together. By holding out examples where concepts are composed during training, we are assured that the model has not seen any of these combinations. The only way to compose these concepts together is through the implicit causal model (via the reduced form SEM (3)) that is learned during training. We evaluate model performance with the SW metric (18), comparing concept compositions from the model against ground-truth hold out compositions.

The same patterns in performance hold here for concept composition as we saw for concept learning in Section 4.1—namely, our context module consistently achieves better performance than baselines and ablations, demonstrating successful concept composition due to the module's intervention mechanism. For example, looking again at the 3DIdent results in Figure 2 (middle), we see that the colour range of both the background and the object are shifted from the orange–green range (■■■■■) to the blue–red range (■■■■■).

These experiments convincingly demonstrate the context module's ability to learn causally disentangled representations through both concept learning and composition. They also highlight the usefulness of the `quad` dataset, where concept compositions are easily simulated, allowing us to replicate multiple training runs and compute $p$-values.

### 4.3 Reconstruction evaluation

As a final validation of the module, Table 3 shows the (validation) reconstruction evaluation of our approach (CM+NVAE) versus NVAE over both interventional datasets—Morpho MNIST and 3DIdent—as well as an observational dataset—CelebA. Since CelebA does not have interventional data, we use this as an important ablation to test how well our module performs when used with *conditional* data (i.e., by conditioning on attributes in CelebA). Additional metrics for comparison against other models on MNIST and CelebA can be found in Vahdat & Kautz (2020, Table 1).

Table 3: Validation reconstruction performance (ELBO loss in bits per dimension).

| Dataset | Method | |
|---|---|---|
| | **CM+NVAE** | **NVAE** |
| **MNIST** | 0.149 | **0.144** |
| **3DIdent** | 0.754 | **0.517** |
| **CelebA** | 2.13 | **2.08** |

We draw two conclusions from these results. First, adding the context module incurs only a minor cost in reconstruction performance, while granting additional concept learning and composition abilities explored in Sections 4.1-4.2. Figure 2 demonstrates that despite this small cost, the perceptual quality of the samples is still very high. Second, conditional rather than interventional data does not significantly alter these conclusions, since on CelebA, where interventional data is not available, CM+NVAE performs only slightly worse than base NVAE.

## 5 Conclusion and Limitations

Designing generative models with structured latent spaces that enable intervention, composition, and OOD generation is an outstanding challenge. Existing approaches either sacrifice structure for flexibility, or flexibility for structure. We argue that such a tradeoff is not necessary and that arbitrarily flexible models can indeed learn useful structure. To this end, we propose a simple decoder-only context module that can be appended to a black-box decoder that learns causally disentangled latent spaces. Our experiments provide both quantitative and qualitative evidence for this. While existing evaluations have focused on reconstruction as a metric to quantify OOD performance, we argue that OOD generation is in fact more appropriate since it also measures the ability of a model to learn useful concepts that can be manipulated at inference time. The context module we propose here enables genuinely OOD, compositional generation without hardcoding prior knowledge or structure into the model. Moreover, it avoids the difficult problem of latent causal discovery. Our experiments with CelebA show that while interventions are not necessary, conditioning alone leads to some loss in fidelity and this also presents a potential direction for future work. While OOD generation is one metric for evaluating genuine OOD generalization, we encourage the community to continue pursuing alternative metrics. Finally, since our module imposes no hard constraints, it may be useful in combination with other modules. Understanding how the context module interacts with other architectures is a natural next step and a promising direction for future work.

### Reproducibility statement

Open-source code and complete instructions for reproducing the results are available at https://after.review. Our context module is implemented as a PyTorch module (Ansel et al., 2024) to more easily facilitate integration with existing methods and future development.

LLM USAGE STATEMENT

We made use of large language models (LLMs) to assist with coding and debugging (Python, Bash, LATEX) and for grammar and diction suggestions to polish the writing. No substantive technical content, results, analyses, or citations were produced by the LLMs. We, the authors, take full responsibility for the content of this submission.

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

# Supplementary material:
# Intervening to learn and compose
# causally disentangled representations

## A  RELATED WORK

Our work is closely related to several parallel lines of work on structured generative models, causal representation learning, linear representations, and OOD generalization. Below we compare and contrast our contributions against this literature.

**Structured generative models**  To provide structure such as hierarchical, graphical, causal, and disentangled structures as well as other inductive biases in the latent space, there has been a trend towards building *structured* generative models that directly impose this structure *a priori*. Early work looked at incorporating fixed, known structure into generative models, such as autoregressive, graphical, and hierarchical structure (Germain et al., 2015; Johnson et al., 2016; Sønderby et al., 2016; Webb et al., 2018; Weilbach et al., 2020; Ding et al., 2021; Mouton & Kroon, 2023). This was later translated into known *causal* structure (Kocaoglu et al., 2017; Markham et al., 2023). When the latent structure is unknown, several techniques have been developed to learn useful (not necessarily causal) structure from data (Li et al., 2018; He et al., 2019; Wehenkel & Louppe, 2021; Kivva et al., 2022; Moran et al., 2022). More recently, based on growing interest in disentangled (Bengio, 2013) and/or causal (Schölkopf et al., 2021) representation learning, methods that learn causal structure have been developed (Markham & Grosse-Wentrup, 2020; Moraffah et al., 2020; Yang et al., 2021; Ashman et al., 2022; Shen et al., 2022; Kaltenpoth & Vreeken, 2023). In contrast to this line of work, which emphasizes hard graphical constraints, we neither learn nor impose any fixed graphical structure. Instead, we emphasize performance on concept learning and composition as downstream tasks.

**Causal disentanglement and representation learning**  Causal representation learning (Schölkopf et al., 2021) is a rapidly developing area that involves, among other goals, two key objectives: 1) Learning disentangled latent factors along with 2) Learning latent causal structure. The former is what we refer to as *causal disentanglement* (see Definition 2.1 and its causal interpretation). The latter problem has seen tremendous theoretical progress in recent years (Brehmer et al., 2022; Shen et al., 2022; Lachapelle et al., 2022; Moran et al., 2022; Kivva et al., 2022; Buchholz et al., 2023; Gresele et al., 2021; Ahuja et al., 2023; von Kügelgen et al., 2023; Morioka & Hyvärinen, 2023; Liu et al., 2023; Mameche et al., 2023; Yao et al., 2023; Varici et al., 2023; Sturma et al., 2023; Xu et al., 2024; Zhang et al., 2024; Li et al., 2024; Varici et al., 2024; Bing et al., 2024; Ahuja et al., 2024; Talon et al., 2024; Yao et al., 2024). Recent work has also pushed in the direction of identifying concepts (Leemann et al., 2023; Rajendran et al., 2024; Fokkema et al., 2025). Our work draws inspiration from these lines of work, which articulate precise conditions under which latent causal discovery is possible *in principle*. By contrast, our focus is on methodological aspects of causal disentanglement *in practice*. While the former is a notoriously difficult task, the latter only needs to exploit learned causal invariances: In particular, causal disentanglement is possible without learning latent causal structure.

**Linear representations**  Generative models are known to represent concepts linearly in embedding space (e.g. Mikolov et al., 2013; Szegedy et al., 2013; Radford et al., 2015); see also Levy & Goldberg (2014); Arora et al. (2016); Gittens et al. (2017); Allen & Hospedales (2019); Ethayarajh et al. (2018); Seonwoo et al. (2019). This phenomenon has been well-documented over the past decade in both language models (Mikolov et al., 2013; Pennington et al., 2014; Arora et al., 2016; Conneau et al., 2018; Tenney et al., 2019; Elhage et al., 2022; Burns et al., 2022; Tigges et al., 2023; Nanda et al., 2023; Moschella et al., 2022; Li et al., 2023; Park et al., 2023; Gurnee et al., 2023; Cunningham et al., 2024; Jiang et al., 2024) and computer vision (Radford et al., 2015; Raghu et al., 2017; Bau et al., 2017; Engel et al., 2017; Kim et al., 2018; Chen et al., 2020; Wang et al., 2023; Trager et al., 2023). Our approach actively exploits this tendency by searching for concept representations as linear projections of the embeddings learned by a black-box model. In contrast to this prior work, our emphasis is on applying these empirical observations for causal disentanglement, compositional generalization, and OOD generation.

**OOD generalization** A growing line of work studies the OOD generalization capabilities of generative models, with the general observation being that existing methods struggle to generalize OOD (Xu et al., 2022; Mo et al., 2024; Montero et al., 2022; 2021; Schott et al., 2022). It is worth noting that most if not all of this work evaluates OOD generalization using reconstruction on held-out OOD samples, as opposed to generation. For example, a traditional VAE may be able to reconstruct held-out samples, but it is not possible to actively sample OOD. See Section 2 for more discussion.

## B IDENTIFIABILITY OF THE PROPOSED MODEL

In this appendix we set up and prove Theorem 4.1.

### B.1 SET-UP

Our setup is the following: Let $\mathbf{x} = (\mathbf{x}_1, \ldots, \mathbf{x}_n), \mathbf{c} = (\mathbf{c}_1, \ldots, \mathbf{c}_{d_c})$ and $\mathbf{e} = (\mathbf{e}_1, \ldots, \mathbf{e}_{d_e})$ be random vectors taking values in $\mathcal{X}, \mathcal{C}$ and $\mathcal{E}$, respectively. For simplicity, we assume that $\mathcal{X} \subseteq \mathbb{R}^n$, $\mathbf{e} \in \mathcal{E} \subseteq \mathbb{R}^{d_e}$, and $\mathbf{x} = f(\mathbf{e})$ for some embeddings $\mathbf{e} \in \mathcal{E}$. We assume that $f$ is injective and differentiable. This map is allowed to be arbitrarily non-linear: We make no additional assumptions on $f$.

The random variables $\mathbf{c}_j$ will be referred to as "concepts": Intuitively they represent abstract concepts such as shape, size, colour, etc. that may not be perfectly represented by any particular embedding in $\mathbf{e}$. The *linear representation hypothesis* (Section A) is an empirical observation that abstract concepts can be approximately represented as linear projections of a sufficiently flexible latent space. This is operationalized by assuming that $\mathbf{c}_j \approx C_j \mathbf{e}$ (see (A1) below). The matrix $C_j$ is referred to as the *representation* of the concept $\mathbf{c}_j$. If $\mathbf{c}_j \in \mathbb{R}$ (i.e. $C_j \in \mathbb{R}^{1 \times d_e}$), then $\mathbf{c}_j$ is referred to as an *atom* and $C_j$ its *atomic representation*. We will often denote atoms by $\mathbf{a}_j$ for notational clarity.

We consider $(\mathbf{x}, \mathbf{c}, \mathbf{e})$ satisfying the following:

(A1) $\mathbf{c}_j = C_j \mathbf{e} + \eta_j$ for all $j \in [d_c]$ for some real matrices $C_1 \ldots, C_{d_c}$, where $\eta_j \sim \mathcal{N}(0, \Lambda_j)$ for some diagonal matrix $\Lambda_j \succ 0$.

(A2) $\mathbf{x} = f(\mathbf{e})$ for some injective and differentiable function $f$.

(A3) There is a DAG $\mathsf{G} = ([d_c], E)$ such that

$$\mathbf{c}_j = \sum_{k \in \mathrm{pa}_\mathsf{G}(j)} \alpha_{kj} \mathbf{c}_k + \boldsymbol{\varepsilon}_j, \quad \text{for all } j \in [d_c] \tag{5}$$

and where $\boldsymbol{\varepsilon}_1, \ldots, \boldsymbol{\varepsilon}_{d_c}$ are mutually independent and $\mathbf{e}, \eta$ are independent of $\boldsymbol{\varepsilon} = (\boldsymbol{\varepsilon}_1, \ldots, \boldsymbol{\varepsilon}_{d_c})$.

Note that these assumptions are not necessary *in practice*—as evidenced by our experiments in Section 4 which clearly violate these assumptions—and are merely used here to provide a proof-of-concept identifiability result based on our architectural choices. The representation (5) implies, in particular, the usual factorization

$$p(\mathbf{c}_1, \ldots, \mathbf{c}_{d_c}) = \prod_{j=1}^{d_c} p(\mathbf{c}_j \mid \mathrm{pa}_\mathsf{G}(j)),$$

which formalizes the notion of causal disentanglement as in Schölkopf et al. (2021).

Our goal is to identify the concept marginal $p(\mathbf{c})$ and concept representations $(C_1, \ldots, C_{d_c})$ from concept interventions. Concept interventions are well-defined through the structural causal model defined by the DAG $\mathsf{G}$ and its structural equations. We consider single-node concept interventions where $\mathbf{c}_j$ is stochastically set to be centered at $\boldsymbol{\mu}_j$ with variance $\Omega_j \succ 0$ (making this precise requires some set-up; see Appendix B.3.1 for details).

By identifiability, we mean the usual notion of identifiability up to permutation, shift, and scaling from the literature. Namely, for any two sets of parameters $(f, C_1, \ldots, C_{d_c})$ and $(\widetilde{f}, \widetilde{C}_1, \ldots, \widetilde{C}_{d_c})$ that

generate the same observed marginal $p(\mathbf{x})$, there exists permutations $P_j$, diagonal scaling matrices $D_j$, and a shift $b \in \mathbb{R}^{d_e}$, such that

$$C_j f^{-1}(\mathbf{x}) = D_j P_j \widetilde{C}_j(\widetilde{f}^{-1}(\mathbf{x}) + b), \quad \text{for all } j \text{ and } \mathbf{x}. \tag{6}$$

Moreover, there exists an invertible linear transformation $L$ such that the concepts satisfy

$$C_j = P_j \widetilde{C}_j L^{-1}. \tag{7}$$

This definition of identifiability, which aligns with similar notions of identifiability that have appeared previously (e.g. Squires et al., 2023; von Kügelgen et al., 2023; Rajendran et al., 2024; Fokkema et al., 2025), implies in particular that the concepts $\mathbf{c}_j$ are identifiable up to permutation, shift, and scale, and that their representations are identifiable up to a linear transformation. As a result, the concept marginal $p(\mathbf{c})$ is also identifiable (again up to permutation, shift, and scale as well). The shift ambiguity can of course be resolved by assuming the concepts are centered (zero-mean), and the scale ambiguity can be resolved by assuming concepts are normalized (e.g. unit norm).

We make the following assumptions, which are adapted from Rajendran et al. (2024) to the present setting. Note that Rajendran et al. (2024) do not consider a structural causal model over $\mathbf{c}$, and thus the notion of a concept intervention there is not well-defined.

**Assumption 1** (Atomic representations)**.** There exists a set of atomic representations $\mathcal{A} = \{\mathbf{a}_1, \ldots, \mathbf{a}_n\}$ of linearly independent vectors $\mathbf{a}_j \in \mathbb{R}^{d_e}$ such that the rows of each representation $C_j$ are in $\mathcal{A}$. We denote the indices of $\mathcal{A}$ that appear as rows of $C_j$ by $S^j$, i.e. $S^j = \{i : \mathbf{a}_i \text{ is a row in } C_j\}$. We assume that $\cup_j S^j = \mathcal{A}$, i.e. every atom in $\mathcal{A}$ appears in some concept representation $C_j$.

Define a matrix $M \in \mathbb{R}^{n \times d_c}$ to track which concepts use which atoms, i.e.

$$M_{ij} = \begin{cases} \frac{1}{\omega_j^2} & \text{if } i \in S^j \\ 0 & \text{otherwise,} \end{cases} \tag{8}$$

where $\omega_j^2$ are the diagonal entries of $\Omega_j$. Similarly, we define a matrix $B \in \mathbb{R}^{n \times d_c}$ given by

$$B_{ij} = \begin{cases} \frac{(\boldsymbol{\mu}_j)_k}{\omega_j^2} & \text{if } i \in S^j \text{ and the } k\text{th row of } C_j \text{ is } \mathbf{a}_i, \\ 0 & \text{otherwise.} \end{cases} \tag{9}$$

The second assumption ensures that the concepts and associated interventional environments are sufficiently diverse.

**Assumption 2** (Concept diversity)**.** For every pair of atoms $\mathbf{a}_i$ and $\mathbf{a}_j$ with $i \neq j$ there is a concept $\mathbf{c}_k$ such that $i \in S^k$ and $j \notin S^k$. Furthermore, $\text{rank}(M) = n$ and there exists $v \in \mathbb{R}^{d_c}$ such that $v^\top M = 0$ and $(v^\top B)_i \neq 0$ for each coordinate $i$.

### B.2 FORMAL STATEMENT OF THEOREM 4.1

With this setup, we can now state and prove a more formal version of Theorem 4.1.

**Theorem B.1.** *Under Assumptions 1-2 and given single-node interventions on each concept $\mathbf{c}_j$, we can identify the representations $C_j$ and the latent concept distribution $p(\mathbf{c})$, in the sense of (6-7).*

In the context of learning disentangled representations, identifying the concept marginal $p(\mathbf{c})$ is particularly relevant, since this implies we also learn certain (in)dependence relationships between the concepts. In other words, if the "true" concept representations are disentangled (i.e. independent), then we identify disentangled concepts.

In relation to existing work, we make the following remarks:

- While our proof ultimately relies on techniques from Rajendran et al. (2024), Theorem B.1 does not trivially follow from their results, which do not cover interventions over the concept marginal $p(\mathbf{c})$. Extending their results to the case of concept interventions, including making sure these interventions are well-defined causal objects, is the main technical difficulty in our proof. This is surprisingly tricky since we do not assume a full structural causal model over $(\mathbf{x}, \mathbf{c}, \mathbf{e})$: Interventions are only well-defined over $\mathbf{c}$, which makes deducing the downstream effects on the observed $\mathbf{x}$ somewhat subtle.

- Various results in CRL (e.g. Buchholz et al., 2023; von Kügelgen et al., 2023; Varici et al., 2024, see Section A for more references) prove identifiability results under interventions, however, these results require interventions directly on the embeddings $\mathbf{e}$, as opposed to the concepts $\mathbf{c}$. Theorem B.1, by contrast, requires only $d_c \ll d_e$ total interventions and in doing so directly addresses the technical challenges associated with intervening directly on concepts as opposed to embeddings.

- Leemann et al. (2023); Fokkema et al. (2025) study concept identifiability when the nonlinear function $f$ is known. We avoid this simplifying assumption (sometimes called *post-hoc* identifiability), and dealing with the potential nonidentifiability of $f$ is another technical complication our setting must deal with. Indeed, in our setting we can only identify the behaviour of $f$ on the concept subspaces defined by $C_j$, which is much weaker than identifying all of $f$ on $\mathcal{E}$. On the other hand, we have left finite-sample aspects to future work.

### B.3 Proof of Theorem 4.1

To prove Theorem 4.1, we begin with some notation. Let $\mathcal{M}$ denote the collection of all distributions $P$ over $(\mathbf{x}, \mathbf{c}, \mathbf{e})$ satisfying (A1)-(A3) for some choice of $C_1, \ldots, C_{d_c}, \eta, \mathsf{G}, f_1, \ldots, f_{d_c}$, and $\varepsilon$. For a fixed $\mathsf{G}$, let

$$\mathcal{M}(\mathsf{G}) := \{P = P(\mathbf{x}, \mathbf{c}, \mathbf{e}) \in \mathcal{M} : P \text{ satisfies (A3) with } \mathsf{G}\}. \tag{10}$$

The model $\mathcal{M}(\mathsf{G})$ is nonempty for any $\mathsf{G}$, as can be seen by letting $\mathbf{e}, \eta$ and $\varepsilon$ be normally distributed and taking all functions to be linear. Moreover, $\mathcal{M} = \cup_{\mathsf{G}} \mathcal{M}(\mathsf{G})$. Finally, let $\mathcal{M}_{\mathbf{c}}(\mathsf{G})$ denote the collection of all marginal distributions over $\mathbf{c}$ induced by the distributions in $\mathcal{M}(\mathsf{G})$.

### B.3.1 Concept Interventions

In practice, the edge structure of the graph $\mathsf{G}$ is unknown, and we do not assume this is known in advance. We work in the setting where we can only observe outcomes of the variables $\mathbf{x}$. However, we allow for the possibility to intervene on the variables $\mathbf{c}_1, \ldots, \mathbf{c}_{d_c}$. We consider consider only interventional distributions with single-node targets $I = \{j\}$ for $j \in [m]$. Note that, by definition of $\mathcal{M}_{\mathbf{c}}(\mathsf{G})$, if $P \in \mathcal{M}_{\mathbf{c}}(\mathsf{G})$ then there is a distribution $\widetilde{P} \in \mathcal{M}(\mathsf{G})$ such that $\widetilde{p}(\mathbf{c}) = p(\mathbf{c})$ for all $\mathbf{c} \in \mathcal{C}$. Although immediate from these definitions, we record this as a lemma for future use:

**Lemma B.2.** *If $P \in \mathcal{M}_{\mathbf{c}}(\mathsf{G})$ then there is a distribution $\widetilde{P} \in \mathcal{M}(\mathsf{G})$ such that $\widetilde{p}(\mathbf{c}) = p(\mathbf{c})$ for all $\mathbf{c} \in \mathcal{C}$.*

For the target $I = \{j\}$ and distribution $P \in \mathcal{M}_{\mathbf{c}}(\mathsf{G})$ we say that $P^j$ is an *interventional distribution* for $j$ and $P$ if in $P^j$ we have the following identities:

$$\mathbf{c}_i = \sum_{k \in \mathrm{pa}_{\mathsf{G}}(i)} \alpha_{ki} \mathbf{c}_k + \varepsilon_i,$$

$$\mathbf{c}_j = \boldsymbol{\mu}_j + \eta_j', \quad \text{where } \eta_j' \sim \mathcal{N}(0, \omega_j^2). \tag{11}$$

That is, the structural equations defining $\mathbf{c}_i$ for $i \neq j$ are equal to those defining $\mathbf{c}_i$ in the observational distribution $P$, while the distribution of $\mathbf{c}_j$ is manipulated to be $\mathcal{N}(\boldsymbol{\mu}_j, \omega_j^2)$, independent of the other concepts.

The following is obvious from our assumptions:

**Lemma B.3.** *For any distribution $P \in \mathcal{M}_{\mathbf{c}}(\mathsf{G})$, the interventional distribution $P^j$ defined by (11) is also an element of $\mathcal{M}_{\mathbf{c}}(\mathsf{G})$, i.e. $P^j \in \mathcal{M}_{\mathbf{c}}(\mathsf{G})$.*

Thus, we have the usual, well-defined notion of intervention over the concepts $\mathbf{c}$. The next step is to translate the effect of these interventions onto $\mathbf{e}$ and $\mathbf{x}$.

The following proposition shows that an interventional distribution $P^j \in \mathcal{M}_{\mathbf{c}}(\mathsf{G})$ for $P \in \mathcal{M}_{\mathbf{c}}(\mathsf{G})$ always extends to a well-defined post-interventional joint distribution over $(\mathbf{x}, \mathbf{c}, \mathbf{e})$ that can be interpreted as an interventional distribution for the appropriate choice of joint observational distribution over $(\mathbf{x}, \mathbf{c}, \mathbf{e})$.

**Proposition B.4.** *Let $P^j$ be an interventional distribution for $j$ and $P \in \mathcal{M}_{\mathbf{c}}(\mathsf{G})$. Then there exists a distribution $\widetilde{P}^j \in \mathcal{M}(\mathsf{G})$ such that*

*1. $\widetilde{p}^j(\mathbf{c}) = p^j(\mathbf{c})$ for all $\mathbf{c} \in \mathcal{C}$,*

*2. $\widetilde{p}^j(\mathbf{e}) = \widetilde{p}(\mathbf{e})$ for all $\mathbf{e} \in \mathcal{E}$,*

*3. If $\widetilde{P}$ is induced by the functional relation $\mathbf{x} = f(\mathbf{e})$, and similarly $\widetilde{P}^j$ with $\mathbf{x} = f^j(\mathbf{e})$ then $f^j = f$.*

*Proof.* Lemma B.3 implies that $P^j \in \mathcal{M}_{\mathbf{c}}(\mathsf{G})$. Thus Lemma B.2 implies the existence of joint distributions $\widetilde{P}, \widetilde{P}^j \in \mathcal{M}(\mathsf{G})$ such that

$$\widetilde{p}(\mathbf{c}) = p(\mathbf{c}) \quad \text{and} \quad \widetilde{p}^j(\mathbf{c}) = p^j(\mathbf{c}) \text{ for all } \mathbf{c} \in \mathcal{C}. \tag{12}$$

(1) follows from the definition of $\widetilde{P}^j \in \mathcal{M}(\mathsf{G})$ for $P^j \in \mathcal{M}_{\mathbf{c}}(\mathsf{G})$ from Lemma B.2 above; i.e. (12) above.

(2) follows from the definition of an interventional distribution and the factorization

$$\widetilde{p}(\mathbf{x}, \mathbf{c}, \mathbf{e}) = \widetilde{p}(\mathbf{x} \mid \mathbf{e})\widetilde{p}(\mathbf{c} \mid \mathbf{e})\widetilde{p}(\mathbf{e}),$$

satisfied by any distribution $P \in \mathcal{M}(\mathsf{G})$. Namely, the interventional distribution $P^j$ satisfies

$$\begin{aligned}
\widetilde{p}^j(\mathbf{x}, \mathbf{c}, \mathbf{e}) &= \widetilde{p}^j(\mathbf{x} \mid \mathbf{e})\widetilde{p}^j(\mathbf{c} \mid \mathbf{e})\widetilde{p}^j(\mathbf{e}), \\
&= \widetilde{p}^j(\mathbf{x} \mid \mathbf{e})p^j(\mathbf{c} \mid \mathbf{e})\widetilde{p}^j(\mathbf{e}),
\end{aligned}$$

according to the definition of $\widetilde{P}$. Since the intervention only perturbs the conditional factors $\widetilde{p}(\mathbf{c}_j \mid \mathbf{c}_{\mathrm{pa}_{\mathsf{G}}(j)}, \mathbf{e})$ in $\widetilde{p}(\mathbf{c} \mid \mathbf{e})$ for $j$, we can always choose $\widetilde{P}$ and $\widetilde{P}^j$ such that $\widetilde{p}^j(\mathbf{e}) = \widetilde{p}(\mathbf{e})$.

(3) follows similarly to (2). Namely, since the intervention only perturbs the conditional factors specified above, we can always choose $f^j = f$. $\qquad\square$

We additionally have the following:

**Proposition B.5.** *Let $I_j := \{j\}$ be the collection of single-node intervention targets, and consider the interventional distributions $P^1, \ldots, P^{d_c}$ for $P \in \mathcal{M}_{\mathbf{c}}(\mathsf{G})$. Then there exists $\widetilde{P}, \widetilde{P}^1, \ldots, \widetilde{P}^{d_c} \in \mathcal{M}(\mathsf{G})$ such that*

$$\widetilde{p}^j(\mathbf{e}) = \widetilde{p}(\mathbf{e}) \qquad \text{for all } j \in [d_c]$$

*and $f^j = f$ for all $j$.*

*Proof.* Fix a choice of $\widetilde{P} \in \mathcal{M}(\mathsf{G})$ for $P$, which specifies a function $f$. For the chosen $\widetilde{P}$, we construct $\widetilde{P}, \widetilde{P}^1, \ldots, \widetilde{P}^{d_c}$ according to Proposition B.4. The result follows by applying Proposition B.4 for every $j$. $\qquad\square$

Proposition B.5 will allow us (below) to define more formally the environments that are used to identify concepts. Intuitively, each environment corresponds to single-node interventions on a single concept $\mathbf{c}_j$, however, we can only observe the effect this intervention has on the observed $\mathbf{x}$. The purpose of Propositions B.4-B.5 are to trace the dependence between $\mathbf{x}$ and $\mathbf{c}$—via the embeddings $\mathbf{e}$—according to the model (A1)-(A3).

Formally, let $I_j = \{j\}$ with $\eta'_j \sim \mathcal{N}(0, \omega_j^2)$ and $\mathbf{c}_j = \boldsymbol{\mu}_j + \eta'_j$ according to (11), i.e. each environment corresponds to a single-node intervention on the $j$th concept $\mathbf{c}_j$. By Proposition B.5, there exist post-intervention joint distributions $\widetilde{P}^j \in \mathcal{M}(\mathsf{G})$ over $(\mathbf{x}, \mathbf{c}, \mathbf{e})$ such that

$$\widetilde{p}^j(\mathbf{e}) = \widetilde{p}(\mathbf{e}) \qquad \text{for all } j \in [d_c]$$

and $f^j = f$ for all $j$. Then the $j$th environment corresponds to sampling $\mathbf{x} = f(\mathbf{e})$ where $\mathbf{e} \sim p_j(\mathbf{e}) := \widetilde{p}^j(\mathbf{e} \mid \mathbf{c}_j = \boldsymbol{\mu}_j)$.

### B.3.2  REDUCTION TO CONCEPT IDENTIFICATION

We begin by computing the log-likelihood ratio between the observational and interventional environments. The idea is that the concept representations are revealed through fluctuations in this likelihood ratio. Using Proposition B.5, we have

$$\log p_0(\mathbf{e}) - \log p_j(\mathbf{e}) \propto \log p_0(\mathbf{e}) - \log \widetilde{p}^j(\mathbf{c}_j = \boldsymbol{\mu}_j \,|\, \mathbf{e}) - \log \widetilde{p}^j(\mathbf{e}) \tag{13}$$

$$\propto -\tfrac{1}{2}(\boldsymbol{\mu}_j - C_j\mathbf{e})^T \Omega_j^{-1}(\boldsymbol{\mu}_j - C_j\mathbf{e}) \tag{14}$$

$$\propto \sum_{i=1}^{n} \frac{1}{2} M_{ij}(\mathbf{a}_i^T\mathbf{e})^2 - B_{ij}\mathbf{a}_i^T\mathbf{e} + O(1). \tag{15}$$

From here, we proceed as in Rajendran et al. (2024) (note that in their notation, $\mathbf{e} = \mathbf{z}$). For completeness, we sketch the main steps here. The basic idea is to analyze the system of $d_c$ equations induced by the log-likelihood ratios for each concept intervention. First, we reduce the system to a standard form by transforming the atoms into the standard basis via a change of coordinates. After this change of coordinates, the system can be rewritten as

$$h_j(\mathbf{e}) := \log p_0(\mathbf{e}) - \log p_j(\mathbf{e}) \propto \sum_{i=1}^{n} \frac{1}{2} M_{ij}\mathbf{e}_i^2 - B_{ij}\mathbf{e}_i + O(1). \tag{16}$$

These functions are convex, which allows us to identify the $|S_T|$ where $S_T = \cup_{j\in T} S_j$ by minimizing the convex function $\sum_{j\in T} h_j(\mathbf{e})$. Then a similar induction argument identifies the matrices $M$ and $B$. Now, given two distinct representations of $p(\mathbf{x})$ via nonlinear mixing functions $f$ and $\widetilde{f}$, define $\varphi = \widetilde{f}^{-1} \circ f$. Let $\mathbf{h} = (h_1, \ldots, h_n)$ and write $\mathbf{e}^c = (\mathbf{e}_1, \ldots, \mathbf{e}_n)$ and $\mathbf{e} = (\mathbf{e}^c, \mathbf{e}^\perp) \in \mathbb{R}^{n\times(d_c-n)}$. Let $\iota^\perp(\mathbf{e}^c) = (\mathbf{e}^c, 0)$, $\pi^c(\mathbf{e}) = \mathbf{e}^c$, and $\varphi^\perp : \mathbb{R}^n \to \mathbb{R}^n$ by $\varphi^\perp(\mathbf{e}^c)_i = \varphi(\mathbf{e}, 0)_i$. Note that via the change of variables formula, we can write $\mathbf{h}(\mathbf{e}) = \mathbf{H}(\mathbf{x})$ for some function $\mathbf{H}$. Then

$$\mathbf{h}(\mathbf{e}^c, 0) = \mathbf{H}(f(\mathbf{e}^c, 0)) = \mathbf{H}(\widetilde{g}(\varphi^\perp(\mathbf{e}^c))) = \mathbf{h}(\varphi^\perp(\mathbf{e}^c)), \tag{17}$$

which is the crucial relation used in the proof of Theorem 1 of Rajendran et al. (2024). From here, identifiability follows from a straightforward but lengthy linear algebraic argument based on the first-order conditions for minimizing $\mathbf{h}$ and the identifiability of $M, B$ proved above. The proof of Theorem B.1 follows.

## C  QUAD: A SEMI-SYNTHETIC BENCHMARK FOR COMPOSITIONAL GENERATION

To provide a controllable, synthetic test bed for evaluating composition in a visual environment, we developed a simple semi-synthetic benchmark, visualized in Figures 3–4. quad is a visual environment defined by 8 concepts:

1. quad1: The colour of the first quadrant;
2. quad2: The colour of the second quadrant;
3. quad3: The colour of the third quadrant;
4. quad4: The colour of the fourth quadrant;
5. size: The size of the center object;
6. orientation: The orientation (angle) of the center object;
7. object: The colour of the center object;
8. shape: The shape of the center object (circle, square, pill, triangle).

With the exception of shape, which is discrete, the remaining concepts take values in $[0, 1]$. This environment has the following appealing properties:

- Composing multiple concepts is straightforward and perceptually distinct;
- Concepts are continuous, which allows a variety of soft and hard interventions;
- Sampling is fast and easy, and requires only a few lines of Python code and no external software;

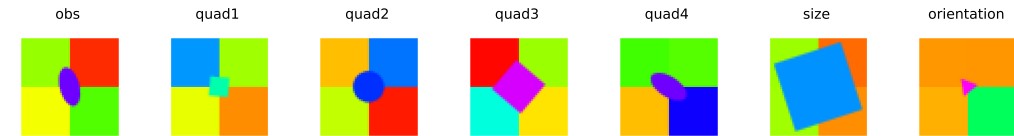

Figure 3: Example images obtained from the `quad` dataset. Each subfigure corresponds to a different context, indicated by the title (`obs` = observational; the others indicate a single-node intervention). For example, in `quad1`, the first (top left) quadrant has been manipulated from orange-green hues to blue-red hues.

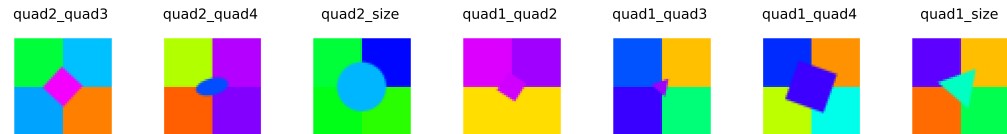

Figure 4: Example images obtained from the `quad` dataset. Each subfigure corresponds to a different double-concept intervention context, indicated by the title. For example, in `quad2_quad3`, the second (top-right) and third (bottom-left) quadrants have been manipulated from orange-green hues to blue-red hues.

- Creating OOD hold-out validation datasets of any size is straightforward;
- This can be simulated on any $n \times n$ grid (as long as $n$ is large enough to distinguish different shapes; $n \geq 16$ is large enough in practice). In our experiments, we used $n = 64$.

These properties make `quad` especially useful for both quantitative and qualitative evaluation; indeed, it was created precisely for this reason. Example images are shown in Figures 3-4.

To describe the experimental setting used in the experiments, let $c_j$ indicate the $j$th concept as listed above. We used seven different contexts, including an observational context and one context each for single-concept interventions on $(c_1, \ldots, c_6)$, i.e. `quad1`, `quad2`, `quad3`, `quad4`, `size`, and `orientation`. We use a sample size of 50 000 images per context. In each context, $c_7$ (`object`) and $c_8$ (`shape`) are sampled uniformly at random. The remaining six concepts were sampled as follows:

- The observational context was generated by randomly sampling $(c_1, \ldots, c_6)$ independently from the range $[0, 0.5]$,
- The interventional contexts were generated by isolating a particular concept and sampling it uniformly from $[0.5, 1]$, while sampling the rest from $[0, 0.5]$.

For example, colours in $[0, 0.5]$ include greens, yellows, and reds, and colours in $[0.5, 1]$ include blues, purples, and magenta. Thus, the object colour takes on any value in every context, whereas the four quadrant colours are restricted in the observational context. When we intervene on a particular quadrant, we see that the colour palette noticeably shifts from greens to blues (Figure 3).

Similarly, we can compose multiple interventions together, as in Figure 4. This facilitates evaluation of the concept composition capabilities of models trained on single interventions, providing a ground truth distribution to compare against generated OOD samples like in Figure 5.

We also made use of the easy-to-generate and perceptually distinct concept compositions in this dateset by evaluating models on held-out contexts (each containing 10 000 images) with double-concept interventions, including `quad2_quad3`, `quad2_quad4`, `quad2_size`, `quad1_quad2`, `quad1_quad3`, `quad1_quad4`, and `quad1_size`. Examples can be seen in Figure 4.

For comparison, see Figure E.1.2 for examples of generated multi-concept compositions (OOD) from a simple 3-layer convolutional model. The training data did not contain any multi-concept compositions.

# D EXPERIMENTAL DETAILS

In this appendix we provide additional details on our experimental protocol and set-up.

## D.1 EVALUATION METRICS

We evaluated each model on the following metrics:

- **Validation ELBO loss**: This is the standard ELBO loss used for validating VAEs. We report this in bits per dimension (bpd), which is the loss scaled by a factor of $\frac{1}{\text{num\_pixels} \cdot \ln 2}$.

- $p$-**Sliced Wasserstein (SW) distance**: Unlike the loss above, which evaluates *individual reconstructed* images with respect to their corresponding ground truth images, this metric (Bonneel et al., 2015) evaluates a *generated distribution* $\mu_{\text{gen}}$ with respect to a ground truth distribution $\mu$. Specifically, we use the implementation of Flamary et al. (2021) to compute a Monte-Carlo approximation of $\mathcal{SWD}_p(\mu_{\text{gen}}, \mu)$, where

$$\mathcal{SWD}_p(\mu, \nu) = \mathop{\mathbb{E}}_{\theta \sim \mathcal{U}(\mathbb{S}^{d-1})} \left( \mathcal{W}_p^p(\theta_{\#}\mu, \theta_{\#}\nu) \right)^{\frac{1}{p}}, \tag{18}$$

and $\theta_{\#}\mu$ stands for the pushforwards of the projection $X \in \mathbb{R}^d \mapsto \langle \theta, X \rangle$ and $p = 2$.

When reporting values for these metrics in tables, we show the mean, averaged over different runs/seeds, followed by the standard error. We generally use a 70/30 training/validation split.

## D.2 DATASETS

In addition to the `quad` benchmark introduced in Appendix C, we used the following benchmark datasets.

**3DIdent** The 3DIdent dataset (Zimmermann et al., 2021) consists of a 3D object, rendered using the Blender rendering engine, under different lighting conditions and orientations. We generated 50 000 samples per context corresponding to single-node interventions on background colour, spotlight colour, and object colour, in addition to the observational context.

**MNIST** We use a variation of the classic MNIST dataset (LeCun et al., 1998) known as Morpho-MNIST (Castro et al., 2019) along with additional affine transformations of MNIST (e.g., as in Simard et al. (2003)). In addition to the observational context (standard MNIST images), the dataset contains the five contexts corresponding to single-concept interventions on concepts `scaled`, `shear`, `swel`, `thic`, and `thin`. The datasets contain 60 000 images per context.

**CelebA** We use the original CelebA data (Liu et al., 2015), using the attributes included in the original dataset to define different contexts. Since we do not have access to interventions on this dataset, we relied on conditioning to create different contexts for different concepts. To minimize leakage between contexts, we chose 6 attributes by minimizing the between-attribute correlations and also choosing attributes that are easily detected visually (e.g. black or blond hair). This resulted in six total contexts of varying sizes, one observational along with single-node interventions on `bags_under_eyes` (41 446 samples), `black_hair` (48 472 samples), `blond_hair` (29 983 samples), `male` (84 434 samples), and `mouth_slightly_open` (97 942 samples).

## D.3 ARCHITECTURES

We tested our context module using two black-box models:

1. (lightweight) $\beta$-VAE: A standard 3-layer convolutional network with latent dimension $\dim(\mathbf{z}) = 128$. This is used for the `quad` experiments in Section 4 and the `quad` and MNIST experiments in Section E.1.

2. NVAE: A deep hierarchical VAE; see Vahdat & Kautz (2020) for details. This is used for the MNIST, 3DIdent, and CelebA experiments in Sections 4 and E.2.

The lightweight-VAE was used for comprehensive ablations and experiments on simple datasets, totalling 160 models overall, with 5 per column per table in Appendix E.1. Since NVAE requires substantially more resources and time to train, this was used to train 8 total models overall, one per column in Table 2 for the 3DIdent data and two per row for the other two datasets in Table 3.

To provide an apples-to-apples baseline, we trained MNIST using both lightweight-VAE and NVAE as base blackbox methods.

### D.4 Additional ablations

We conducted additional ablations to stress test our model and isolate the effect of the context module. This included additional experiments on different models, regularization, and expressivity.

**Models** We ran ablation different base VAEs augmented with the context-module (denoted CM+*VAE)—this is our module attached to the black-box, making full use of the different contexts. To evaluate the proposed module, we compared its performance to three natural baselines:

1. **pooled CM**: CM+*VAE but trained with a single context on the full *pooled* dataset—this is our module attached to the black-box given all the data but not making use of context-specific information, providing an ablation that circumvents our intervention layer (Section 3.2) while maintaining the same architectural complexity.

2. **obs *VAE**: Base VAE (without the context module) trained only on a single observational context—this is the black-box when denied all (non-observational) context information and data.

3. **pooled *VAE**: Base VAE (without the context module) trained with a single context on the full pooled dataset—this is the black-box when given all data but no explicit context information.

Note that due to differences in the size of the pooled vs. observational-only datasets, loss values from obs *VAE are not directly comparable to the others, but are included anyway for reference. Results for $\beta$VAE as the base applied to `quad` and MNIST are shown in Section 4 and Appendix E.1. Results for NVAE as the base applied to 3DIdent are shown in Section 4.

**Regularization** We run CM+$\beta$VAE on MNIST, independently trying out two different regularizers, varying the regularization weight $\lambda$ for each: group lasso and $\ell_2$. These are applied to the weights of the reduced-form SEM $A_0$ in 2. We also try varying $\beta$ (as in $\beta$-VAE). Results are shown in Appendix E.1.3.

**Expressivity** We run CM+$\beta$VAE on MNIST, jointly varying three parameters that control expressiveness of our context module: expressive input width $w_{\exp}$ (which controls latent dimension $\dim(\mathbf{z})$), depth of the expressive layer $h_{\exp}$, and concept width ($w_c = \dim(\mathbf{c}_j) = w_\varepsilon = \dim(\boldsymbol{\varepsilon}_j)$ as in Section 3.2). Results are shown in Appendix E.1.4.

## E Additional results

In this section, we present complete results, including experimental settings described in Appendix D for our context module with the lightweight-VAE black-box architecture as well as additional results on MNIST, 3DIdent, and CelebA for our context module with NVAE.

### E.1 lightweight-VAE

#### E.1.1 Ablations (MNIST and quad)

A description of the architecture and experiment is given in Appendix D. Table 4 shows ablation results after 200 epochs of training on MNIST, and Table 5 is after 1000 epochs.[1] These are comparable, but while Table 5 has shows slightly degraded reconstruction evaluation, it also shows

---

[1]All other experiments in Appendix E.1 are accordingly run with no ablation and on 200 epochs.

better concept leaning and in-distribution generalization. Table 1 shows ablation results on `quad` after 200 epochs.

Altogether, these ablations studies suggest that the cost of the structured representation learned by the context module is only a slight degradation in some of the usual numerical metrics (like validation ELBO), while the SW metrics and Figure 5 show sensible ID concept learning and OOD concept composition using our module.

Table 4: Summary of ablation results for reconstruction and concept learning on MNIST (200 epochs).

| | Method | | | |
|---|---|---|---|---|
| | CM+$\beta$VAE | pooled CM | obs $\beta$VAE | pooled $\beta$VAE |
| **Reconstruction** (validation ELBO $\downarrow$) | | | | |
| | $0.295 \pm 0.009$ | $0.289 \pm 0.010$ | $\mathbf{0.183} \pm 0.001$ | $0.236 \pm 0.005$ |
| **Concept Learning** (SW $\downarrow$) | | | | |
| **obs** | $0.094 \pm 0.009$ | $0.087 \pm 0.010$ | $\mathbf{0.041} \pm 0.001$ | $0.093 \pm 0.010$ |
| **scaled** | $\mathbf{0.065} \pm 0.006$ | $0.131 \pm 0.017$ | $0.173 \pm 0.006$ | $0.117 \pm 0.011$ |
| **shear** | $0.123 \pm 0.006$ | $0.109 \pm 0.006$ | $\mathbf{0.091} \pm 0.002$ | $0.100 \pm 0.004$ |
| **shift** | $0.136 \pm 0.006$ | $0.112 \pm 0.005$ | $\mathbf{0.092} \pm 0.002$ | $0.106 \pm 0.004$ |
| **swel** | $0.141 \pm 0.004$ | $0.120 \pm 0.015$ | $\mathbf{0.081} \pm 0.004$ | $0.141 \pm 0.006$ |
| **thic** | $0.231 \pm 0.007$ | $0.238 \pm 0.016$ | $\mathbf{0.184} \pm 0.008$ | $0.267 \pm 0.011$ |
| **thin** | $\mathbf{0.056} \pm 0.006$ | $0.086 \pm 0.014$ | $0.122 \pm 0.006$ | $0.074 \pm 0.008$ |
| **mean** | $0.121 \pm 0.022$ | $0.126 \pm 0.020$ | $\mathbf{0.112} \pm 0.020$ | $0.128 \pm 0.024$ |

Table 5: Summary of ablation results for reconstruction and concept learning on MNIST (1000 epochs).

| | Method | | | |
|---|---|---|---|---|
| | CM+$\beta$VAE | pooled CM | obs $\beta$VAE | pooled $\beta$VAE |
| **Reconstruction** (validation ELBO $\downarrow$) | | | | |
| | $0.315 \pm 0.008$ | $0.314 \pm 0.012$ | $\mathbf{0.189} \pm 0.002$ | $0.239 \pm 0.009$ |
| **Concept Learning** (SW $\downarrow$) | | | | |
| **obs** | $0.062 \pm 0.007$ | $0.097 \pm 0.022$ | $\mathbf{0.037} \pm 0.001$ | $0.068 \pm 0.008$ |
| **scaled** | $\mathbf{0.063} \pm 0.007$ | $0.137 \pm 0.028$ | $0.174 \pm 0.003$ | $0.148 \pm 0.013$ |
| **shear** | $0.096 \pm 0.006$ | $0.116 \pm 0.010$ | $\mathbf{0.091} \pm 0.005$ | $0.091 \pm 0.004$ |
| **shift** | $0.128 \pm 0.008$ | $0.123 \pm 0.012$ | $\mathbf{0.090} \pm 0.002$ | $0.096 \pm 0.004$ |
| **swel** | $0.113 \pm 0.011$ | $0.138 \pm 0.027$ | $\mathbf{0.078} \pm 0.001$ | $0.114 \pm 0.016$ |
| **thic** | $\mathbf{0.155} \pm 0.011$ | $0.223 \pm 0.029$ | $0.199 \pm 0.009$ | $0.215 \pm 0.014$ |
| **thin** | $\mathbf{0.068} \pm 0.010$ | $0.094 \pm 0.019$ | $0.127 \pm 0.005$ | $0.103 \pm 0.015$ |
| **mean** | $\mathbf{0.098} \pm 0.014$ | $0.132 \pm 0.016$ | $0.114 \pm 0.021$ | $0.119 \pm 0.018$ |

Table 6: Summary of ablation results for reconstruction, concept learning, and compositional generalization on the `quad` dataset.

| | Method | | | |
|---|---|---|---|---|
| | **CM+$\beta$VAE** | **pooled CM** | **obs $\beta$VAE** | **pooled $\beta$VAE** |
| **Reconstruction** (validation ELBO ↓) | | | | |
| | $7.663 \pm 1.260$ | $10.440 \pm 2.241$ | $\mathbf{2.329} \pm 0.025$ | $2.909 \pm 0.259$ |
| **Concept Learning** (SW ↓) | | | | |
| **obs** | $\mathbf{0.032} \pm 0.006$ | $0.084 \pm 0.012$ | $0.043 \pm 0.001$ | $0.087 \pm 0.013$ |
| **orientation** | $\mathbf{0.036} \pm 0.005$ | $0.082 \pm 0.010$ | $0.045 \pm 0.002$ | $0.086 \pm 0.013$ |
| **quad1** | $\mathbf{0.045} \pm 0.004$ | $0.257 \pm 0.017$ | $0.241 \pm 0.014$ | $0.209 \pm 0.009$ |
| **quad2** | $\mathbf{0.047} \pm 0.002$ | $0.240 \pm 0.009$ | $0.247 \pm 0.009$ | $0.223 \pm 0.008$ |
| **quad3** | $\mathbf{0.046} \pm 0.002$ | $0.243 \pm 0.016$ | $0.230 \pm 0.012$ | $0.220 \pm 0.009$ |
| **quad4** | $\mathbf{0.039} \pm 0.001$ | $0.238 \pm 0.008$ | $0.238 \pm 0.010$ | $0.210 \pm 0.011$ |
| **size** | $\mathbf{0.059} \pm 0.003$ | $0.164 \pm 0.009$ | $0.150 \pm 0.004$ | $0.113 \pm 0.005$ |
| **mean** | $\mathbf{0.043} \pm 0.003$ | $0.187 \pm 0.029$ | $0.170 \pm 0.035$ | $0.164 \pm 0.025$ |
| **Composition** (SW ↓) | | | | |
| **(quad1, orientation)** | $\mathbf{0.081} \pm 0.019$ | $0.253 \pm 0.010$ | $0.248 \pm 0.011$ | $0.226 \pm 0.018$ |
| **(quad1, quad2)** | $\mathbf{0.145} \pm 0.027$ | $0.363 \pm 0.010$ | $0.300 \pm 0.015$ | $0.284 \pm 0.007$ |
| **(quad1, quad3)** | $\mathbf{0.147} \pm 0.039$ | $0.327 \pm 0.016$ | $0.343 \pm 0.017$ | $0.305 \pm 0.008$ |
| **(quad1, quad4)** | $\mathbf{0.097} \pm 0.014$ | $0.324 \pm 0.018$ | $0.337 \pm 0.011$ | $0.299 \pm 0.023$ |
| **(quad1, size)** | $\mathbf{0.142} \pm 0.027$ | $0.257 \pm 0.017$ | $0.259 \pm 0.016$ | $0.222 \pm 0.008$ |
| **(quad2, orientation)** | $\mathbf{0.074} \pm 0.018$ | $0.236 \pm 0.013$ | $0.212 \pm 0.007$ | $0.209 \pm 0.014$ |
| **(quad2, quad3)** | $\mathbf{0.162} \pm 0.021$ | $0.347 \pm 0.019$ | $0.361 \pm 0.015$ | $0.311 \pm 0.017$ |
| **(quad2, quad4)** | $\mathbf{0.139} \pm 0.018$ | $0.305 \pm 0.013$ | $0.324 \pm 0.007$ | $0.293 \pm 0.012$ |
| **(quad2, size)** | $\mathbf{0.129} \pm 0.019$ | $0.274 \pm 0.021$ | $0.274 \pm 0.005$ | $0.205 \pm 0.014$ |
| **mean** | $\mathbf{0.124} \pm 0.011$ | $0.298 \pm 0.015$ | $0.295 \pm 0.017$ | $0.261 \pm 0.015$ |

### E.1.2 EXAMPLES OF CONCEPT COMPOSITION (QUAD)

Figure 5 depicts samples generated from the context module appended to the lightweight-VAE described in Appendix D.3.

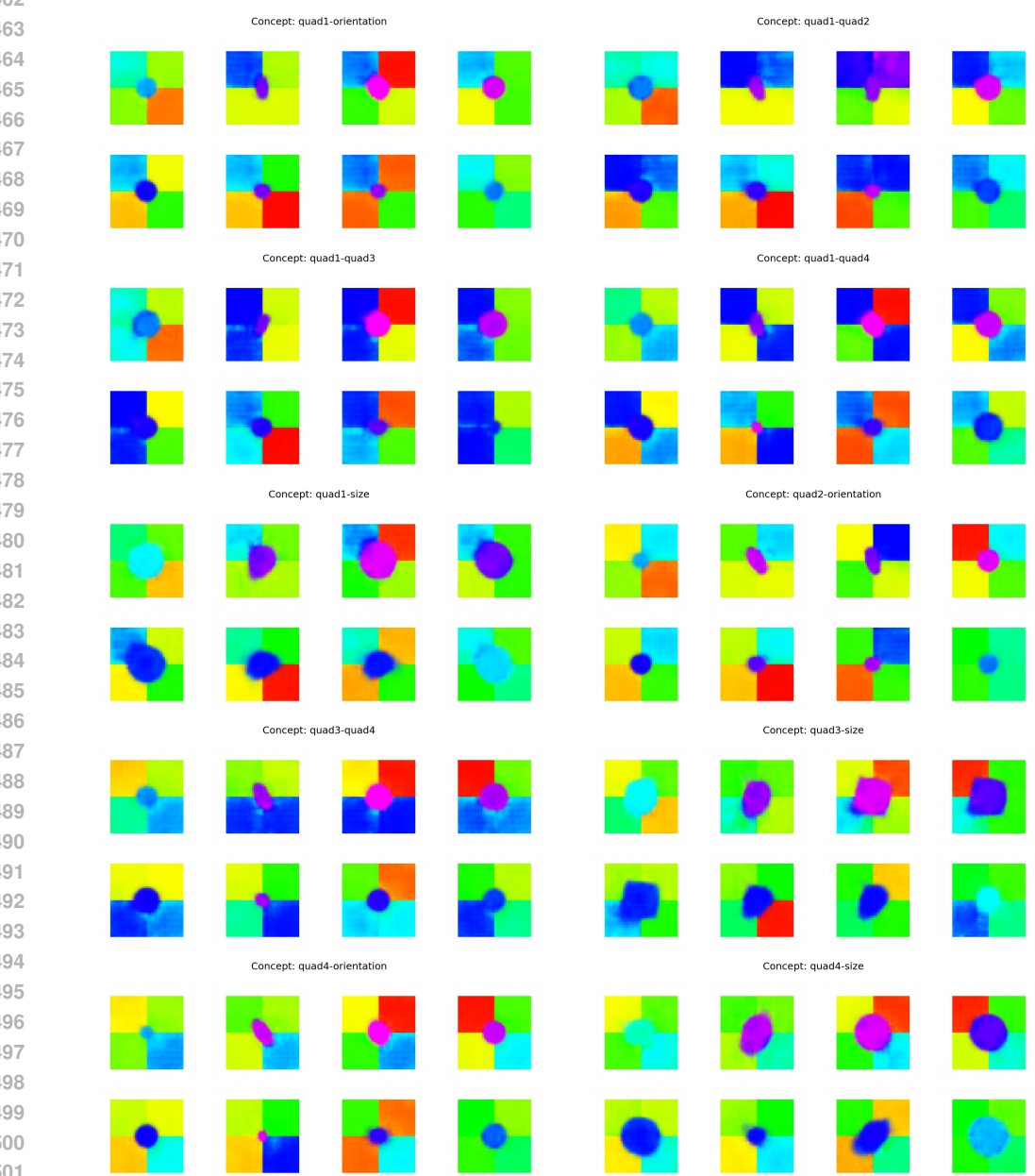

Figure 5: Example generated OOD images from a run of quad.

### E.1.3 REGULARIZATION (MNIST)

A description of the architecture and experiment is given in Appendix D. Tables 7 and 8 respectively show results of group lasso (Yuan & Lin, 2006) and $\ell_2$ regularization, while Table 9 shows the effects regularizing the latent space of the VAE by varying $\beta$ (Higgins et al., 2017).[2] Altogether, these results indicate relatively stable performance across regularization weights while suggesting that tuning

---

[2]All other experiments in Appendix E.1 are run with $\lambda = 0$ for group lasso and $\ell_2$ and with $\beta = 1$.

these regularizers could be worthwhile (e.g., notice that $\lambda = 1$ improves performance compared to other lambda values as well as compared to no regularization in Tables 4 and 5).

Table 7: Group lasso regularization on MNIST

| | $\lambda = 1$ | $\lambda = 10$ | $\lambda = 100$ |
|---|---|---|---|
| **Reconstruction** (validation ELBO ↓) | | | |
| | $0.298 \pm 0.005$ | $0.291 \pm 0.008$ | $\mathbf{0.287} \pm 0.006$ |
| **Concept Learning** (SW ↓) | | | |
| **obs** | $\mathbf{0.064} \pm 0.005$ | $0.071 \pm 0.006$ | $0.070 \pm 0.006$ |
| **scaled** | $0.070 \pm 0.021$ | $0.059 \pm 0.009$ | $\mathbf{0.059} \pm 0.006$ |
| **shear** | $\mathbf{0.079} \pm 0.004$ | $0.095 \pm 0.009$ | $0.087 \pm 0.005$ |
| **shift** | $0.094 \pm 0.009$ | $0.118 \pm 0.012$ | $\mathbf{0.093} \pm 0.011$ |
| **swel** | $\mathbf{0.088} \pm 0.007$ | $0.106 \pm 0.011$ | $0.104 \pm 0.004$ |
| **thic** | $\mathbf{0.179} \pm 0.008$ | $0.195 \pm 0.019$ | $0.213 \pm 0.015$ |
| **thin** | $0.087 \pm 0.011$ | $\mathbf{0.072} \pm 0.012$ | $0.086 \pm 0.008$ |
| **mean** | $\mathbf{0.095} \pm 0.004$ | $0.102 \pm 0.004$ | $0.102 \pm 0.003$ |

Table 8: $\ell_2$ regularization on MNIST

| | $\lambda = 1$ | $\lambda = 10$ | $\lambda = 100$ |
|---|---|---|---|
| **Reconstruction** (validation ELBO ↓) | | | |
| | $0.308 \pm 0.014$ | $\mathbf{0.292} \pm 0.006$ | $0.297 \pm 0.008$ |
| **Concept Learning** (SW ↓) | | | |
| **obs** | $\mathbf{0.063} \pm 0.006$ | $0.076 \pm 0.008$ | $0.079 \pm 0.010$ |
| **scaled** | $\mathbf{0.061} \pm 0.013$ | $0.072 \pm 0.018$ | $0.064 \pm 0.010$ |
| **shear** | $\mathbf{0.082} \pm 0.005$ | $0.100 \pm 0.014$ | $0.091 \pm 0.006$ |
| **shift** | $0.104 \pm 0.008$ | $0.110 \pm 0.010$ | $\mathbf{0.102} \pm 0.010$ |
| **swel** | $\mathbf{0.090} \pm 0.012$ | $0.111 \pm 0.015$ | $0.107 \pm 0.013$ |
| **thic** | $\mathbf{0.168} \pm 0.006$ | $0.200 \pm 0.010$ | $0.200 \pm 0.025$ |
| **thin** | $0.082 \pm 0.014$ | $\mathbf{0.068} \pm 0.014$ | $0.069 \pm 0.004$ |
| **mean** | $\mathbf{0.093} \pm 0.004$ | $0.105 \pm 0.005$ | $0.102 \pm 0.005$ |

Table 9: Latent space regularization (as in $\beta$-VAE) on MNIST

| | $\beta = 0.5$ | $\beta = 2$ | $\beta = 4$ |
|---|---|---|---|
| **Reconstruction** (validation ELBO ↓) | | | |
| | $\mathbf{0.259} \pm 0.006$ | $0.340 \pm 0.008$ | $0.372 \pm 0.008$ |
| **Concept Learning** (SW ↓) | | | |
| **obs** | $\mathbf{0.068} \pm 0.007$ | $0.092 \pm 0.007$ | $0.098 \pm 0.006$ |
| **scaled** | $\mathbf{0.051} \pm 0.002$ | $0.070 \pm 0.006$ | $0.087 \pm 0.009$ |
| **shear** | $\mathbf{0.094} \pm 0.006$ | $0.101 \pm 0.004$ | $0.118 \pm 0.009$ |
| **shift** | $\mathbf{0.112} \pm 0.012$ | $0.116 \pm 0.009$ | $0.138 \pm 0.009$ |
| **swel** | $\mathbf{0.113} \pm 0.009$ | $0.127 \pm 0.006$ | $0.132 \pm 0.007$ |
| **thic** | $\mathbf{0.202} \pm 0.014$ | $0.216 \pm 0.013$ | $0.219 \pm 0.010$ |
| **thin** | $\mathbf{0.056} \pm 0.007$ | $0.070 \pm 0.008$ | $0.081 \pm 0.004$ |
| **mean** | $\mathbf{0.099} \pm 0.003$ | $0.113 \pm 0.003$ | $0.125 \pm 0.003$ |

### E.1.4 Expressivity (MNIST)

A description of the architecture and experiment is given in Appendix D. Tables 10 and 11 show varying widths for an expressive layer depth respectively of 2 and 5: the tuples in the column headers indicate $(w_{\exp}, w_c)$.[3] These results suggest relatively stable/robust results across different configurations while the best performance is obtained by greater expressive layer depth $h_{\exp} = 5$, high expressive layer input width $w_{\exp} = 50$, and a significant bottleneck with a concept width of $w_c = 5$.

Table 10: Expressivity with depth $h_{exp} = 2$ on MNIST as $(w_{exp}, w_c)$ varies

|  | (15,22) | (22,22) | (22,5) | (50,22) | (50,5) |
|---|---|---|---|---|---|
| **Reconstruction** (validation ELBO ↓) | | | | | |
|  | $0.378 \pm 0.010$ | $0.389 \pm 0.012$ | $\mathbf{0.296} \pm 0.003$ | $0.366 \pm 0.009$ | $0.307 \pm 0.010$ |
| **Concept Learning** (SW ↓) | | | | | |
| **obs** | $0.076 \pm 0.002$ | $0.089 \pm 0.010$ | $0.083 \pm 0.008$ | $0.076 \pm 0.004$ | $\mathbf{0.070} \pm 0.009$ |
| **scaled** | $0.089 \pm 0.014$ | $0.092 \pm 0.021$ | $0.076 \pm 0.012$ | $\mathbf{0.062} \pm 0.005$ | $0.067 \pm 0.006$ |
| **shear** | $0.094 \pm 0.003$ | $0.101 \pm 0.010$ | $0.096 \pm 0.009$ | $0.104 \pm 0.007$ | $\mathbf{0.091} \pm 0.013$ |
| **shift** | $0.150 \pm 0.011$ | $0.143 \pm 0.015$ | $\mathbf{0.107} \pm 0.013$ | $0.150 \pm 0.004$ | $0.110 \pm 0.016$ |
| **swel** | $0.108 \pm 0.006$ | $0.133 \pm 0.025$ | $0.115 \pm 0.014$ | $\mathbf{0.093} \pm 0.006$ | $0.112 \pm 0.015$ |
| **thic** | $0.163 \pm 0.009$ | $0.226 \pm 0.011$ | $0.219 \pm 0.010$ | $\mathbf{0.159} \pm 0.013$ | $0.189 \pm 0.023$ |
| **thin** | $0.088 \pm 0.009$ | $0.093 \pm 0.011$ | $\mathbf{0.062} \pm 0.006$ | $0.077 \pm 0.004$ | $0.080 \pm 0.015$ |
| **mean** | $0.110 \pm 0.003$ | $0.125 \pm 0.006$ | $0.108 \pm 0.004$ | $\mathbf{0.103} \pm 0.003$ | $0.103 \pm 0.006$ |

Table 11: Expressivity with depth $h_{exp} = 5$ on MNIST as $(w_{exp}, w_c)$ varies

|  | (15,22) | (15,5) | (22,22) | (22,5) | (50,22) | (50,5) |
|---|---|---|---|---|---|---|
| **Reconstruction** (validation ELBO ↓) | | | | | | |
|  | $0.342 \pm 0.009$ | $0.315 \pm 0.011$ | $0.378 \pm 0.018$ | $\mathbf{0.303} \pm 0.006$ | $0.370 \pm 0.021$ | $0.316 \pm 0.012$ |
| **Concept Learning** (SW ↓) | | | | | | |
| **obs** | $0.070 \pm 0.004$ | $0.079 \pm 0.005$ | $0.078 \pm 0.006$ | $0.072 \pm 0.007$ | $0.119 \pm 0.035$ | $\mathbf{0.063} \pm 0.004$ |
| **scaled** | $0.064 \pm 0.004$ | $\mathbf{0.052} \pm 0.008$ | $0.060 \pm 0.005$ | $0.053 \pm 0.004$ | $0.082 \pm 0.021$ | $0.062 \pm 0.014$ |
| **shear** | $0.099 \pm 0.005$ | $0.101 \pm 0.008$ | $0.101 \pm 0.010$ | $0.089 \pm 0.005$ | $0.137 \pm 0.035$ | $\mathbf{0.079} \pm 0.004$ |
| **shift** | $0.116 \pm 0.012$ | $0.112 \pm 0.008$ | $0.127 \pm 0.015$ | $0.120 \pm 0.006$ | $0.154 \pm 0.032$ | $\mathbf{0.092} \pm 0.010$ |
| **swel** | $0.096 \pm 0.004$ | $0.121 \pm 0.008$ | $0.106 \pm 0.005$ | $0.107 \pm 0.006$ | $0.151 \pm 0.035$ | $\mathbf{0.094} \pm 0.005$ |
| **thic** | $0.188 \pm 0.009$ | $0.185 \pm 0.011$ | $0.202 \pm 0.015$ | $0.181 \pm 0.011$ | $0.227 \pm 0.053$ | $\mathbf{0.173} \pm 0.013$ |
| **thin** | $0.121 \pm 0.009$ | $\mathbf{0.065} \pm 0.008$ | $0.110 \pm 0.009$ | $0.083 \pm 0.007$ | $0.099 \pm 0.020$ | $0.100 \pm 0.010$ |
| **mean** | $0.108 \pm 0.003$ | $0.102 \pm 0.003$ | $0.112 \pm 0.004$ | $0.101 \pm 0.003$ | $0.138 \pm 0.013$ | $\mathbf{0.095} \pm 0.003$ |

## E.2 Additional NVAE Results

### E.2.1 Additional NVAE ablation results on 3DIdent

The raw results used to construct Table 2 in the main text are shown here in Table 12.

### E.2.2 Additional examples of concept composition

Figure 6 contains additional examples of composing concepts from the NVAE-based context module. Each row corresponds to actual generated samples from the trained model in different contexts:

1. The first row shows generated observational samples;

2. The second and third row contain generated single-node interventions;

3. The final row shows generated samples where two distinct concepts are composed together.

---

[3]All other experiments in Appendix E.1 are run with $h_{\exp} = 2$, $w_{\exp} = 15$, and $w_c = 5$.

Table 12: Results on validation ELBO metric for reconstruction and SW metric (lower is better) on concept learning and composition performance across 3DIdent contexts.

| | | Method | | | |
|---|---|---|---|---|---|
| | | **CM+NVAE** | **pooled CM** | **obs NVAE** | **pooled NVAE** |
| | **val ELBO** | 0.754 | 2.293 | 1.581 | **0.517** |
| Concept learning | **obs** | **0.019** | 0.024 | 0.053 | 0.028 |
| | **object** | **0.017** | 0.065 | 0.089 | 0.058 |
| | **background** | **0.018** | 0.059 | 0.066 | 0.053 |
| | **spotlight** | **0.018** | 0.022 | 0.045 | 0.025 |
| | **mean** | **0.018** | 0.043 | 0.063 | 0.041 |
| | **$p$-value** | — | 0.059 | 0.009 | 0.038 |
| Composition | **(obj, bg)** | **0.042** | 0.075 | 0.085 | 0.068 |
| | **(bg, sl)** | **0.046** | 0.069 | 0.095 | 0.067 |
| | **(obj, sl)** | **0.056** | 0.073 | 0.083 | 0.063 |
| | **mean** | **0.048** | 0.072 | 0.088 | 0.066 |
| | **$p$-value** | — | 0.018 | 0.012 | 0.046 |

For MNIST and 3DIdent, the final row of compositional generations is genuinely OOD in that the training data does not contain any examples where both concepts have been intervened upon. Recall that CelebA is trained using conditional as opposed to interventional samples as an ablation on the sensitivity to non-interventional data.

Additional examples of both concept learning (sampling after single-target interventions) and composition (sampling after multi-target interventions) are provided in Appendices E.2.3–E.2.5.

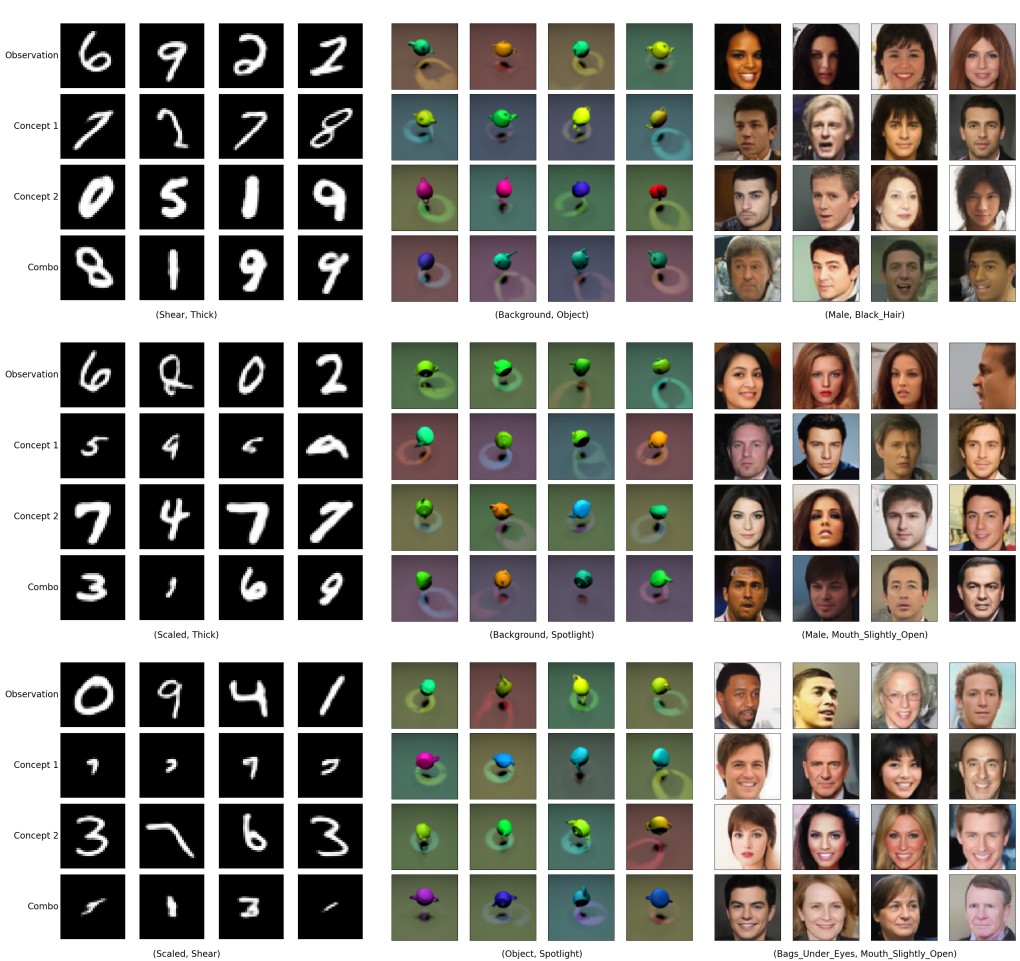

Figure 6: Additional examples of concept composition in MNIST (left), 3DIdent (middle), and CelebA (right). In MNIST and 3DIdent, the samples are OOD: The training data did not contain any examples with these concepts composed together. The CelebA results are an ablation to understand the effect of conditioning vs intervention, and so there is some leakage between concepts, where the interventional datasets (MNIST, CelebA) show no leakage.

### E.2.3 ADDITIONAL MNIST FIGURES

See Figure 7 for concept learning and Figure 8 for composition.

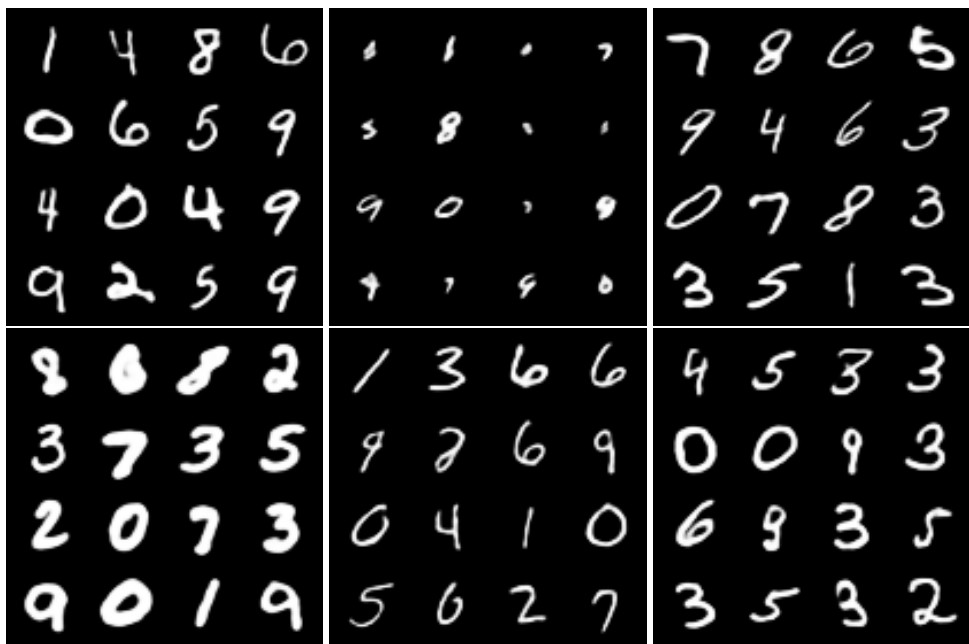

Figure 7: Examples of learned concepts (sampling after single-target intervention) in MNIST. (top) observational, scaled, shear (bottom) thic, thin, swel.

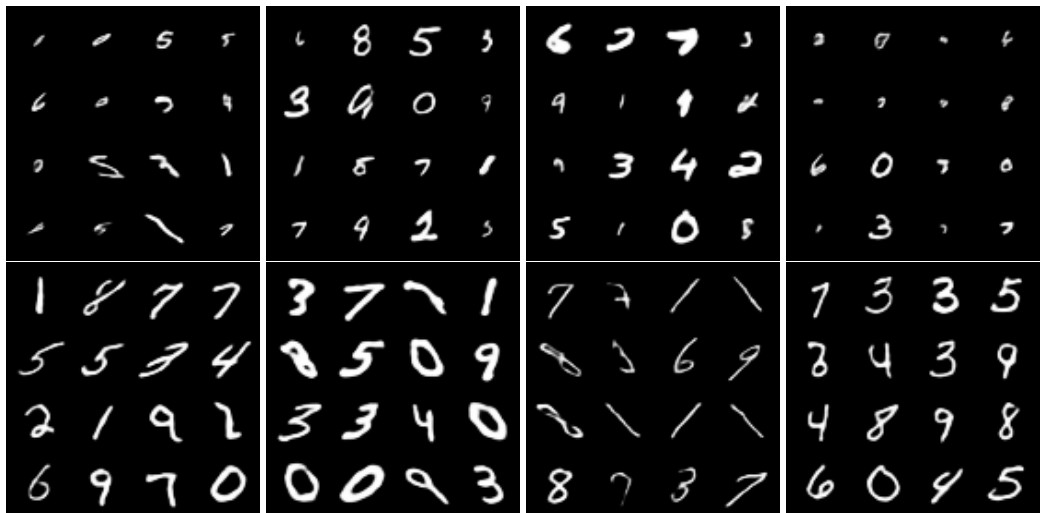

Figure 8: Examples of OOD concept composition (sampling after multi-target intervention) in MNIST.

### E.2.4 ADDITIONAL 3DIDENT FIGURES

See Figure 9 for concept learning and Figure 10 for composition.

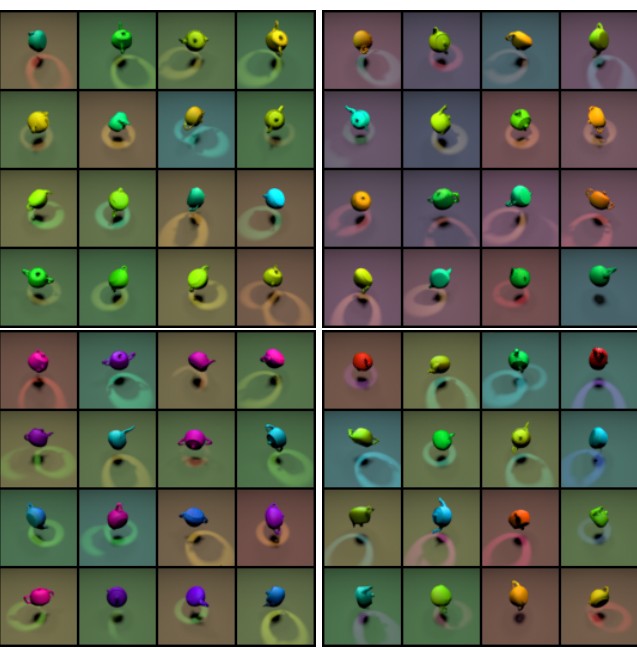

Figure 9: Examples of learned concepts (sampling after single-target intervention) in 3DIdent. (top) observational, background (bottom) object, spotlight.

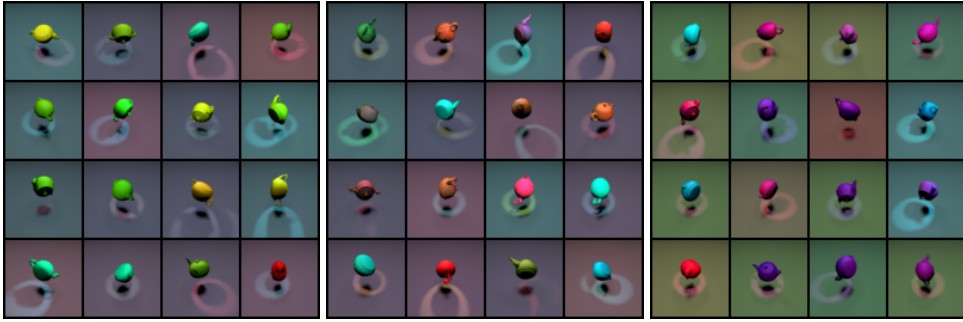

Figure 10: Examples of OOD concept composition (sampling after multi-target intervention) in 3DIdent.

### E.2.5 ADDITIONAL CELEBA FIGURES

See Figure 11 for concept learning and Figure 12 for composition.

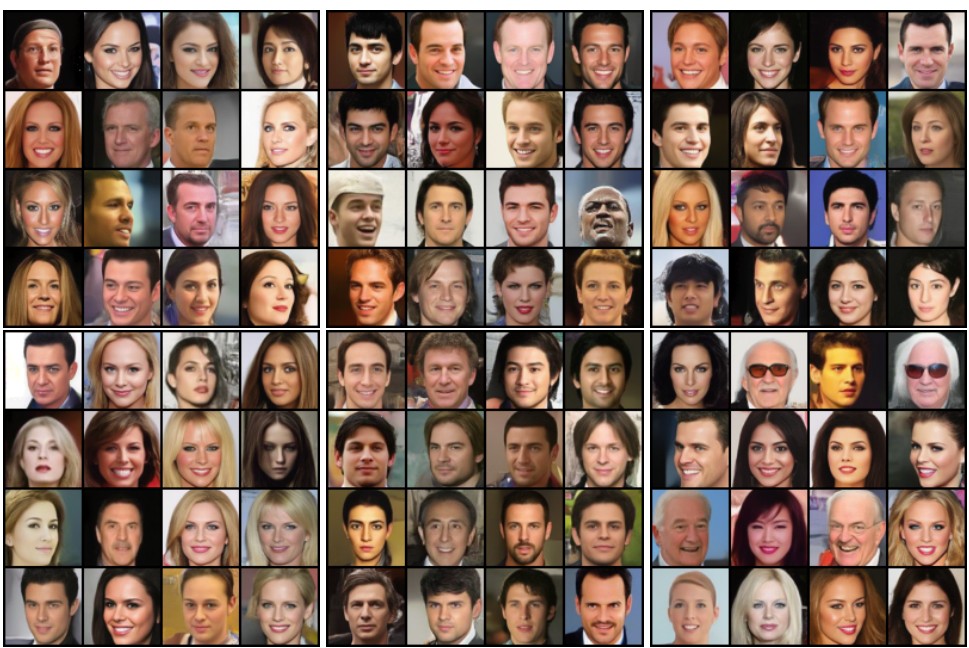

Figure 11: Examples of learned concepts (sampling after single-target intervention) in CelebA. (top) observational, bags under eyes, black hair (bottom) blond hair, male, mouth slightly open.

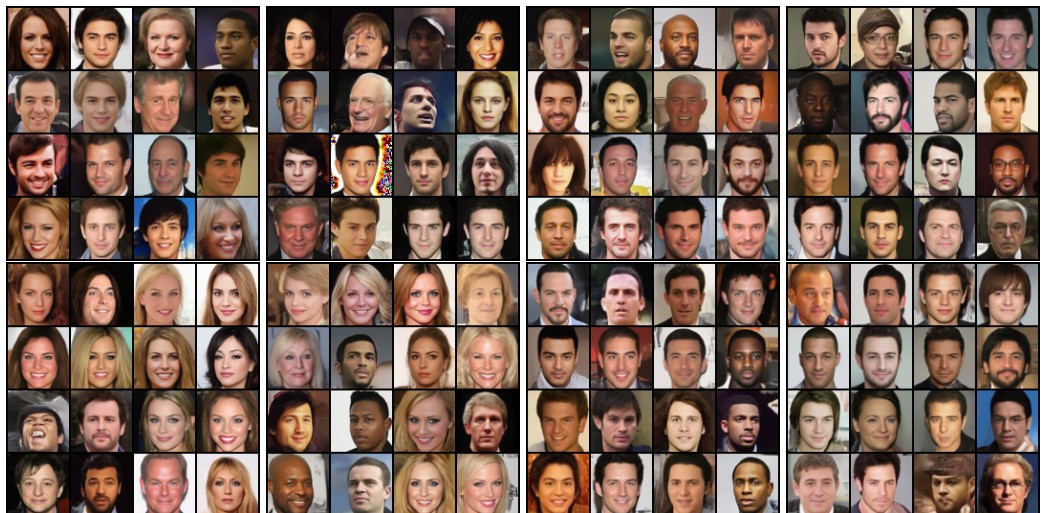

Figure 12: Examples of OOD concept composition (sampling after multi-target intervention) in CelebA.

