# OpenReview forum: "Intervening to learn and compose causally disentangled representations"
_ICLR.cc/2026/Conference — ICLR 2026 Conference Withdrawn Submission_

### Official Review · Reviewer_K2Xc · 2025-10-16

**Soundness:** 2
**Presentation:** 1
**Contribution:** 1
**Rating:** 2
**Confidence:** 4

**Summary:**

The paper adds a simple decoder module that helps complex generative models learn causal and composable latent factors for out-of-distribution generation.

**Strengths:**

1. I like the idea not assuming independent factors and instead keeping a reduced-form SCM inside the module.

2. Plug-in, decoder-only model and easy to use for black box encoder decoder structure.

**Weaknesses:**

1. The problem formalization/setup part is not clear (lines 161–237).
The authors should clarify the formalization of the problem setup and assumptions in the main text. Since this paper aims to contribute both theoretically and empirically, explicitly expressing the definitions and assumptions in more rigors way would make the presentation much clearer.

For example, you assume c = C e, but it is unclear whether this means that for a specific dataset with fixed concepts, all possible encoders (and their embeddings) are assumed to satisfy this property, or whether you only assume that there exists some encoder such that c = C e (∃ e).

Another example: in line 212, the phrase “as a result” lacks logical justification for moving from c = C e to a linear SEM. The equation c = C e does not imply that the internal relationships among the components of c are linear or that they can be represented as a linear SEM. I think c = C e and the linear SEM should be treated as two parallel assumptions rather than one deriving from the other.

2. Lines 263–269 are also confusing.
In Remark 3.1, you say “Due to the reduced-form SEM above, our approach does not and cannot model the structural causal model encoded by α_kj,” but when explaining how you perform interventions, you repeatedly mention “setting α·j = 0,” “zero out the row α·j,” etc. I suggest rephrasing the Concept Interventions section in the main text (perhaps adding pseudocode or a clear figure) to make this crucial part easier for readers to understand.
By the way, “we zero out the row α·j while replacing the column α_j· with a new column β_j·” — here “column” and “row” might be a typo.

3. Most of the experiments appear to be conditional generation/controllable generation rather than true interventions.
This is becaus the factors in the datasets used for experiments seem to be independent. For example, in the appendix:
''The remaining six concepts were sampled as follows:
• The observational context was generated by randomly sampling (c₁, …, c₆) independently from [0, 0.5];
• The interventional contexts were generated by isolating a particular concept and sampling it uniformly from [0.5, 1], while sampling the rest from [0, 0.5].  ''

In these cases, I do not see a significant difference between your claimed “intervention” and conditional/controllable generation. Introducing an SCM to represent independent factors would be clearly redundant and unnecessary. For a paper titled “Intervention …,” I would expect to see real interventions where changing one factor modifies the distributions of its downstream factors according to a causal order.

4. It would be better if the paper included a comparison with other works on OOD generation that propose lightweight modules or regularization terms easily integrated into existing encoder–decoder architectures (if have any).

**Questions:**

See Weaknesses.

---

### Official Review · Reviewer_NfYz · 2025-10-30

**Soundness:** 3
**Presentation:** 3
**Contribution:** 3
**Rating:** 6
**Confidence:** 3

**Summary:**

The paper proposes a plug-in “context module” that turns any black-box VAE decoder into a causally disentangled generator. Concepts are modeled as linear projections of the original latent code; a reduced-form SEM with learnable intervention slices is appended to the decoder. Single- and multi-concept interventions can then be performed at test time without ever seeing the corresponding combinations during training. A new semi-synthetic benchmark (quad) is introduced for controlled evaluation. Theoretical justification is provided via an identifiability result under single-node interventions.

**Strengths:**

1. Minimal architectural commitment: The module is small, differentiable, and works on top of existing pre-trained VAEs without fine-tuning the encoder/decoder.
2. Strong empirical gains: CM+NVAE achieves ≈ 2× lower sliced-Wasserstein distance on OOD compositions vs. pooled baselines on quad, 3DIdent, MNIST, CelebA.
3. Novel evaluation protocol: Shifts the focus from OOD reconstruction to OOD generation, a strictly harder task that better reflects causal disentanglement.
4. Identifiability guarantee: First result to show that single-node concept interventions (not embeddings) suffice to recover linear concept representations up to scale/permutation, without assuming a known causal graph.
5. Reproducibility: Code, hyper-parameters, and the quad dataset are publicly provided.

**Weaknesses:**

1. Limited concept granularity: All experiments assume 1-D scalar concepts; extension to vectorial or hierarchical concepts is mentioned but not evaluated.
2. Intervention design: Interventions are synthetic (uniform shift in colour/scale); real-world interventions (e.g. medical imaging artefacts) may violate linearity or Gaussian exogeneity assumptions.
3. Missing baselines: No comparison with recent weakly-supervised disentanglement methods (e.g. CausalVAE, Ada-GVAE) or energy-based OOD generators; only β-VAE and NVAE are used.
4. Reconstruction penalty : CM+NVAE loses ≈ 0.2 bits/dim on 3DIdent (Table 3); acceptability for high-fidelity domains (faces, video) is not discussed.
5. Theoretical gaps：Identifiability proof assumes injective, differentiable f; real VAE decoders are piece-wise linear (ReLU) and rarely injective over the entire prior support.

**Questions:**

1. How does the method behave if the encoder’s original latent space is not approximately linear in the chosen concepts?
2. Can the SEM tensor be factorised or made low-rank to scale to high-dimensional concepts?
3. Why not compare with CausalVAE or Disentangled GANs that also perform interventions?
4. What happens if interventions are correlated (e.g. colour & texture change together), violating the independent exogenous noise assumption?

---

### Official Review · Reviewer_XtZU · 2025-11-01

**Soundness:** 2
**Presentation:** 3
**Contribution:** 2
**Rating:** 4
**Confidence:** 2

**Summary:**

This paper proposes a context module (CM) — a decoder-only extension for existing encoder-decoder models (e.g., VAE, NVAE) — to learn causally disentangled representations and enable OOD compositional generation. The idea is elegant and supported by theory, but the empirical evidence does not fully support the claim of resolving the expressivity–structure tradeoff. Reconstruction quality drops notably, scalability is unclear, and dependence on interventional data limits practical utility.

**Strengths:**

+ Novel and modular design: A practical decoder-only component that adds causal structure to standard generative models.

+ OOD focus: The paper evaluates that generation rather than reconstruction is conceptually strong.

+ Theory and experiments: This work provides an identifiability proof and well-structured experiments with meaningful ablations.

**Weaknesses:**

+ Poor scalability: The intervention layer grows rapidly with the number of concepts; experiments are limited to small-scale setups.

+ Dependence on interventional data: Results degrade on CelebA, suggesting limited applicability to observational datasets.

**Questions:**

1. Does the approach extend to diffusion or transformer-based decoders?

2. How does the CM scale to larger concept sets (e.g., >10)?

3. How much computational overhead does the context module add?

---

### Note · Authors · 2025-12-02

I have read and agree with the venue's withdrawal policy on behalf of myself and my co-authors.